

# Validation of OMI, GOME-2A and GOME-2B tropospheric NO₂, SO₂ and HCHO products using MAX-DOAS observations from 2011 to 2014 in Wuxi, China

Yang Wang[1], Steffen Beirle[1], Johannes Lampel[1,2], Mariliza Koukouli[3], Isabelle De Smedt[4], Nicolas Theys[4], Ang Li[5], Dexia Wu[5], Pinhua Xie[5,6,7], Cheng Liu[8,6,5], Michel Van Roozendael[4] and Thomas Wagner[1]

[1] Max Planck Institute for Chemistry, Mainz, Germany
[2] Institute of Environmental Physics, University of Heidelberg, Heidelberg, Germany
[3] Laboratory of Atmospheric Physics, Aristotle University of Thessaloniki, Thessaloniki, Greece
[4] Belgian Institute for Space Aeronomy (BIRA-IASB), Brussels, Belgium
[5] Anhui Institute of Optics and Fine Mechanics, Chinese Academy of Sciences, Hefei, China
[6] CAS Center for Excellence in Urban Atmospheric Environment, Institute of Urban Environment, Chinese Academy of Sciences, Xiamen, China
[7] School of Environmental Science and Optoeclectronic Technology, University of Science and Technology of China, Hefei, China
[8] School of Earth and Space Sciences, University of Science and Technology of China, Hefei, China

*Correspondence to*: Yang Wang (y.wang@mpic.de), Ang Li (Angli@aiofm.ac.cn), Cheng Liu (chliu81@ustc.edu.cn)

**Abstract.**

Tropospheric vertical column densities (VCDs) of NO₂, SO₂ and HCHO derived from Ozone Monitoring Instrument (OMI) on AURA and Global Ozone Monitoring Experiment 2 aboard METOP-A (GOME-2A) and METOP-B (GOME-2B) are widely used to characterize the global distributions, trends, dominating sources of the trace gases and for comparisons with chemical transport models (CTM). We use tropospheric VCDs and vertical profiles of NO₂, SO₂ and HCHO derived from MAX-DOAS measurements from 2011 to 2014 in Wuxi, China, to validate the corresponding products derived from OMI, GOME-2A/B by different scientific teams (daily and bimonthly averaged data). Prior to the comparison we investigate the effects of the spatial and temporal coincidence criteria for MAX-DOAS and satellite data on the comparison results. We find that the distance of satellite data from the location of the MAX-DOAS station is the dominating effect, and we make suggestions for the spatial (20km for OMI NO₂ and SO₂ products and 50km for OMI HCHO and all GOME-2A/B products) and temporal averaging (2 hours around satellite overpass time). We also investigate the effect of clouds on both MAX-DOAS and satellite observations. Our results indicate that the discrepancies between satellite and MAX-DOAS results increase with increasing effective cloud fractions and are dominated by the cloud effect on the satellite products. Our comparison results indicate a systematic underestimation of all SO₂ (40% to 57%) and HCHO products (about 20%) and an overestimation of the GOME-2A/B NO₂ products (about 30%) (DOMINO NO₂ product is only slightly underestimated by 1%). To better understand the reasons for the differences, we recalculated the AMFs for satellite observations based on the shape factors (SFs) derived from MAX-DOAS. The recalculated satellite VCDs agree better with the MAX-DOAS VCDs





than those from the original products by up to 10%, 47% and 35% for $NO_2$, $SO_2$ and HCHO, respectively. The improvement is strongest for periods with large trace gas VCDs. Finally we investigate the effect of aerosols on the satellite retrievals. We find an increasing underestimation of the OMI $NO_2$, $SO_2$ and HCHO products with increasing AOD by up to 8%, 12% and 2%, respectively. One reason for this finding is that aerosols systematically affect the satellite cloud retrievals and can lead to apparent effective cloud fractions of up to 10% and apparent cloud top pressures of down to 830 hPa for the typical urban region in Wuxi. We show that in such cases the implicit aerosol correction could cause a strong underestimation of tropospheric VCDs by up to about 45%, 77% and 100% for $NO_2$, $SO_2$ and HCHO, respectively. For such conditions it might be better to apply AMFs for clear sky conditions than AMFs based on the satellite cloud retrievals.

We find that the satellites systematically overestimate the magnitude of the diurnal variations of $NO_2$ and HCHO. No significant weekly cycle for all trace gases is found by either the satellites or the MAX-DOAS measurements.

# 1 Introduction

Nitrogen oxides ($NO_x \equiv NO_2 + NO$), sulphur dioxide ($SO_2$), and formaldehyde (HCHO) play critical roles in the tropospheric chemistry through various gas phase and multi-phase chemical reactions (Seinfeld and Pandis, 1998). In an urban and industrialized region, anthropogenic emissions from traffic, domestic heating, factories, power plants and biomass burning significantly elevate the concentrations of these (and other) trace gases (TGs) in the boundary layer (Environmental Protection Agency, 1998; Seinfeld and Pandis, 1998). There is strong evidence that aerosol particles formed through photochemistry of nitrogen oxides, $SO_2$ and VOCs significantly contribute to haze pollution events occurring frequently around megacities and urban agglomerations in China, like the Jing–Jin–Ji region and the Yangtze River Delta region (Crippa et al., 2014; Huang et al., 2014; Jiang et al., 2015; Fu et al., 2014). The aerosols also impact the local radiative forcing through direct (e.g. McCormic and Ludwig, 1967) and indirect effects (Lohmann and Feichter, 2005). Understanding global and regional distributions and temporal variations of the TGs, and further identifying and quantifying their dominant sources can provide a firm basis for a better understanding of the formation mechanisms of haze pollution and for the development of mitigation strategies.

Since 1995 a series of sun-synchronous satellites, such as ERS-2, ENVISAT, AURA, METOP-A and METOP-B, were launched carrying UV/vis/NIR spectrometers with moderate spectral resolution, which allowed scientists to determine the global distribution of several important tropospheric trace gases including $NO_2$, HCHO and $SO_2$ for the first time. The first instrument was the Global Ozone Monitoring Experiment (GOME) (Burrows et al., 1999), followed by the SCanning Imaging Absorption spectroMeter for Atmospheric CHartographY (SCIAMACHY) (e.g. Bovensmann et al., 1999), the Ozone Monitoring Instrument (OMI) (Levelt et al., 2006a, b), and the GOME-2A and GOME-2B instruments (Callies et al., 2000; Munro et al., 2006, 2016). The OMI and GOME-2A/B instruments are still in operation. A large number of studies developed retrieval algorithms to acquire the tropospheric vertical column densities (VCD) of $NO_2$ (e.g. Boersma et al., 2004, 2007 and 2011; Richter et al., 2005; Beirle et al., 2010 and Valks et al., 2011), $SO_2$ (e.g. Krueger et al., 1995; Eisinger and



Burrows, 1998; Carn et al., 2004; Krotkov et al., 2006; Richter et al., 2006 and 2009; Yang et al., 2007; Lee et al., 2009; Nowlan et al., 2011; Rix et al., 2012; Li et al., 2013; Theys et al., 2015) and HCHO (Chance et al., 2000; Palmer et al., 2001; Wittrock et al., 2006a; De Smedt et al., 2008, 2012 and 2015; Kurosu, 2008; Millet et al., 2008; Hewson et al., 2013; González Abad et al., 2015) for all the satellite instruments. In this validation study we include several products, which are

published recently and widely used: for $NO_2$ the near-real-time OMI DOMINO v2.0 (Boersma et al., 2007 and 2011) and the GOME-2A/B TM4NO2A (Boersma et al., 2004); for $SO_2$ the operational OMSO2 OMI product (Li et al., 2013) published by National Aeronautics and Space Administration (NASA), the O3M-SAF operational GOME-2A product published by the German Aerospace Centre (DLR) (Rix et al., 2012 and Hassinen et al., 2016) and the OMI, GOME-2A/B products developed by BIRA (Theys et al., 2015); for HCHO the OMI and GOME-2A/B products developed by BIRA (De Smedt et

al., 2008, 2012 and 2015). Many users already benefit from these products for several atmospheric applications, e.g. detection and quantification of emissions, identification of transport processes and chemical transformations, and for the comparison with model simulations (e.g. Beirle et al., 2003 and 2011; Martin et al., 2003; Richter et al., 2005; van der A et al., 2008; Herron-Thorpe et al., 2010; Gonzi et al., 2011 and Barkley et al., 2012).

Although several studies have paid efforts to improve the satellite retrievals, still significant differences compared to ground

based measurements were reported by several validation studies, e.g. a systematic underestimation of the tropospheric VCDs of $NO_2$, $SO_2$ and HCHO was obtained for OMI by > 30% in or near Beijing, China (Ma et al., 2013; Theys, et al., 2015 and De Smedt et al., 2015; Jin, et al., 2016). The satellite retrieval errors are mainly attributed to the slant column retrievals, the stratospheric correction (for $NO_2$) and the tropospheric air mass factor (AMF) calculations. The AMF uncertainties are related to several factors, such as the surface albedo, the cloud and aerosol properties, the a-priori (relative) profile (also

referred to as shape factor (SF) in the following) as well as interpolation errors of the discrete look-up table entries (Lin et al., 2014). Thus validation studies for satellite products using independent ground-based measurements are essential to quantify uncertainties, identify dominant error sources and to further improve the satellite retrieval algorithms.

Since about 15 years, the Multi Axis - Differential Optical Absorption Spectroscopy (MAX-DOAS) technique (Hönninger and Platt, 2002; Bobrowski et al., 2003; Van Roozendael et al., 2003; Hönninger et al., 2004; Wagner et al., 2004 and

Wittrock et al., 2004), is applied to retrieve tropospheric vertical profiles of TGs and aerosols from spectra of scattered UV/Visible sunlight measured at different elevation angles (e.g. Frieß et al., 2006, 2011 and 2016; Wittrock et al., 2006b; Irie et al., 2008 and 2011; Clemer et al., 2010; Li et al., 2010 and 2012; Vlemmix et al., 2010, 2011 and 2015b; Wagner et al., 2011; Yilmaz, 2012; Hartl and Wenig, 2013 and Wang et al., 2013a and b). MAX-DOAS observations provide valuable information that can be applied for the quantification of air pollutants (e.g. Li et al., 2012; Hendrick et al., 2014; Wang et al.,

2014a; Wang et al., 2016) and for the validation of tropospheric satellite products (e.g. Irie et al., 2012 and 2016; Ma et al., 2013; Kanaya et al., 2014; Theys, et al., 2015 and De Smedt et al., 2015; Jin et al., 2016) and results of chemical transport model (CTM) simulations (e.g. Vlemmix et al., 2015a). The tropospheric vertical profiles are also valuable for the evaluation of SFs used in the satellite AMF calculations. Here it is important to note that errors of the tropospheric AMFs usually dominate the systematic errors of tropospheric satellite products especially in highly polluted (especially urban and industrial)



regions (Boersma et al., 2011; Theys et al., 2015 and De Smedt et al., 2015), but studies on the effect of the SF on the satellite retrievals are still rare. In this study the effect of the SF on the tropospheric AMF will be investigated using the vertical profiles of the TGs derived from the MAX-DOAS observations in Wuxi, China from 2011 to 2014 (Wang et al., 2016).

Wuxi is located about 130 km north-west of Shanghai belonging to the most industrialized part of the Yangtze River delta (YRD) region. YRD including Shanghai City and four nearby provinces is the largest economic region in China and heavily industrialized and can be considered the largest metropolitan area in Asia with the population of about 150 millions. The air pollution due to strong anthropogenic pollutant emissions in this region threatens the health of the inhabitants and has been of great concern in the atmospheric and environmental science community as well as for the public. Several studies already

used satellite products of the pollutants to quantify the corresponding emissions (Ding et al., 2015; Han et al., 2015; Bauwens et al., 2016) in this region. However validation studies for the satellite products in this region are still sparse. Chen et al. (2009), Irie et al. (2012), Kanaya et al. (2014) and Chan et al. (2015) validate the satellite $NO_2$ tropospheric VCD products using MAX-DOAS (or zenith-sky DOAS) measurements in Rudong, Hefei and Shanghai. So far there are no validation reports for $SO_2$ and HCHO products in the YRD region. However several validation studies have been carried out

in other regions of China (e.g. Theys et al., 2015, De Smedt et al., 2015 and Jin et al., 2016).

In this study we validate daily (2 hours around the satellite overpass time) and bi-monthly averaged tropospheric VCDs of $NO_2$, $SO_2$ and HCHO derived from OMI and GOME-2 using the MAX-DOAS observations in Wuxi. To minimise the influence of different air masses detected by MAX-DOAS and satellite instruments, coincidence criteria should be used for both data sets. In this study we investigate the influence of the temporal and spatial coincidence criteria. So far only few

studies (Ma et al., 2013 and Jin et al., 2016) evaluated the cloud effect on the tropospheric TG products. Thus in this study the comparisons for daily average data are performed for different effective cloud fraction (eCF) intervals (Stammes et al., 2008; Wang et al., 2008). Because clouds could also impact the MAX-DOAS results, it is necessary to evaluate the cloud effects on MAX-DOAS and satellite products separately. This issue will also be discussed in this study. We also investigate the weekly cycles and ratios of morning and afternoon values (representing diurnal cycles) observed by the satellite

instruments

Aerosol information is not considered in the AMF calculation for most tropospheric satellite products (one exception is the OMI $NO_2$ product (POMINO) provided by the Peking University over China (Lin et. al., 2014)), but recently such aerosol effects have drawn more and more attention. Shaiganfar et al. (2011), Ma et al. (2013), and Kanaya et al. (2014) found negative biases of the OMI tropospheric $NO_2$ VCDs between 26 and 50 % over areas with high aerosol pollution through the

validation by MAX-DOAS observations. But aerosol effects on the satellite retrievals are still not well understood. Leitão et al. (2010) performed simulation studies and compared the satellite $NO_2$ AMFs for clear sky with different aerosol scenarios. They found that the influence of aerosols on the satellite AMFs depends mainly on the relative vertical distributions of aerosols and TGs. Recently several studies reported that the OMI and GOME-2 cloud retrievals (eCF and cloud top pressure (CTP)) are indeed sensitive to the presence of (strong loads of) aerosols (Boersma et. al., 2011; Lin et al., 2014; Wang et, al.,





2015; Chimot et al., 2016). They also claimed that for some cases of heavy aerosol loads the cloud correction can (partly) account for the aerosol effects on the satellite AMFs (referred to as implicit aerosol correction). For example, Castellanos et al. (2015) demonstrated that for biomass burning aerosols extending to high altitudes (about 2 km), the implicit correction can well correct the aerosol effect on the OMI tropospheric $NO_2$ product. Here it is important to note that the aerosol around

the heavily polluted urban region typically resides close to the surface, showing often an overlap with the trace gas profiles. Elevated aerosol layers can e.g. occur if long range transport, e.g. from biomass burning contributes to the local aerosol load (Wang et al., 2016). Simulation studies by Lin et al., 2014 and Chimot et al., 2016 showed that the impact of the implicit correction is quite dependent on the vertical profiles of aerosols and the TGs. Thus, in many cases, the implicit correction might even increase the errors of the AMF. In this study the tropospheric aerosol extinction profiles acquired from MAX-

DOAS measurements are used to evaluate the aerosol effects on the satellite observations (not only for $NO_2$, but also for $SO_2$ and HCHO) around heavily polluted urban regions.

The paper is organized as follows: in section 2 we describe MAX-DOAS observations in Wuxi and the satellite products involved in this study. We also discuss the cloud effect on the MAX-DOAS results. In section 3 we compare $NO_2$, $SO_2$ and HCHO VCDs derived from MAX-DOAS with those from the satellite instruments. We investigate in particular the impact of

the coincidence criteria and the effects of clouds, SFs and aerosols on the satellite retrievals. In section 4 the conclusions are given.

## 2 MAX-DOAS measurements and satellite data sets

### 2.1 MAX-DOAS in Wuxi

### 2.1.1 MAX-DOAS instrument and data analysis

A MAX-DOAS instrument developed by Anhui Institute of Optics and Fine Mechanics (AIOFM) (Wang et al., 2015 and 2016) is located on the roof of a 11-story building in Wuxi City (Fig. 1 a-1), China (31.57 °N, 120.31 °E, 50 m a.s.l.) and operated by the Wuxi CAS Photonics Co. Ltd from May 2011 to Dec 2014. Wuxi City is located in the YRD region which is typically affected by high loads of $NO_2$, $SO_2$ and HCHO (Fig. 1 a-2, a-3, a-4). The DOAS method (Platt and Stutz, 2008) and the PriAM profile inversion algorithm (Wang et al., 2013a/b and 2016) are applied to derive the vertical profiles of aerosol

extinction (AEs) and volume mixing ratios (VMRs) of $NO_2$, $SO_2$ and HCHO from scattered UV/visible sunlight recorded by the MAX-DOAS instrument at five elevation angles (5 °, 10 °, 20 °, 30 ° and 90 °). The telescope of the instrument is pointed to the north. The data analysis and the results derived from the MAX-DOAS measurements are already described in our previous study (Wang et al., 2016). In that study we also compared the MAX-DOAS results with collocated independent techniques including an AERONET sun photometer, a visibility meter and a long path DOAS. The comparisons were done

for different cloud conditions as derived from a cloud classification scheme based on the MAX-DOAS observations (Wagner et al., 2014 and Wang et al., 2015). One important conclusion of that study is that meaningful trace gas profiles can be





retrieved not only for clear skies, but also for most cloudy conditions (except heavily fog or haze and optically thick clouds). Thus here we use all MAX-DOAS trace gas profiles obtained for these sky conditions (Wang et al., 2016). Here it is important to note that differently from previous studies (e.g. Ma et al., 2013; Jin et al., 2016), we derive the tropospheric VCDs of the TGs by an integration of the vertical profiles, but not by the so-called geometric approximation (e.g. Brinksma

et al., 2008). Our previous study (Wang et al., 2016) demonstrated that the tropospheric trace gas VCDs from the full profile inversion are in general much more accurate than those from the geometric approximation, for which the errors can be up to 30% depending on geometries of sun and measurements, and scenarios of aerosols and TGs.

### 2.1.2 Cloud effect on MAX-DOAS tropospheric VCDs around the satellite overpass time

In the validation procedure the MAX-DOAS VCDs are averaged over a time period of ±one hour around the satellite

overpass time. Typically about ten MAX-DOAS elevation sequences are recorded during that period, during which the cloud conditions can change. This effect is probably most important for the presence of broken cloud cover. Thus in order to evaluate the cloud effect on MAX-DOAS results, we compare the average MAX-DOAS VCDs derived from all measurements in ±1 hour around the satellite overpass time with those from the measurements under clear sky conditions only. Sky conditions are derived from MAX-DOAS measurements (Wang et al., 2015). The OMI overpass time of 13:30

local time (LT) is selected for the investigation of this effect, and similar features are expected for observations around the GOME-2 overpass time. Fig. 2a, b and c show scatter plots and linear regressions of the average MAX-DOAS VCDs from all the measurements in ±1 hour around the satellite overpass time against those under clear sky conditions for $NO_2$, $SO_2$ and HCHO, respectively. Almost 1:1 linear regression lines and correlation coefficients ($R^2$) (the Pearson's product moment correlation coefficient is applied in this paper) close to unity are found for all three species. To quantify the systematic

differences of the TG VCDs, the corresponding mean differences (and standard deviations) are displayed for eCF<10% and eCF>10%, respectively. In general larger standard deviations are found for all three species for eCF>10%, indicating that larger deviations are related to larger eCF. Mean differences of $0.15 \times 10^{15}$ molecules cm$^{-2}$, $0.02 \times 10^{15}$ molecules cm$^{-2}$ and $0.05 \times 10^{15}$ molecules cm$^{-2}$ (corresponding to 0.8%, 0.05% and 0.4% of the average VCDs) are found for $NO_2$, $SO_2$ and HCHO, respectively, indicating that the cloud effect on MAX-DOAS results is probably negligible for the satellite

validations. Here it should be noted that the shown comparison results represent only situations, for which clear and cloudy conditions occur during the two-hour period around the satellite overpass time. Thus we cannot rule out that the errors for measurements under continuous cloud cover are larger. However situations of continuous cloud cover are not relevant for this validation study, because for such conditions no meaningful satellite results can be obtained.

### 2.2 $NO_2$, $SO_2$ and HCHO products derived from OMI

The OMI instrument (Levelt et al., 2006a, b) aboard the sun-synchronous EOS Aura satellite was launched in July 2004. It achieves daily global coverage with a spatial resolution of $24 \times 13$ km$^2$ in nadir and $68 \times 14$ km$^2$ at the swath edges. The overpass time is around 13:30 LT. In this study, we validate the operational level 2 (Boersma et al., 2007 and 2011)





tropospheric $NO_2$ VCD (DOMINO version 2) obtained from the TEMIS website (http://www.temis.nl). The $NO_2$ SCDs are retrieved in the 405–465 nm spectral window using a DOAS algorithm and are converted to $NO_2$ tropospheric VCDs using tropospheric AMFs from a look-up table, which is generated using the DAK radiative transfer model (RTM) (Stammes, 1994), after the stratospheric column was subtracted. SFs of $NO_2$ for the AMF simulations are obtained from the TM4 CTM

(Williams et al., 2009) for individual measurements and can be downloaded from the TEMIS website. TM4 assimilations run at a resolution of $2° \times 3°$ (lat $\times$ lon) and 35 vertical levels up to 0.38 hPa and are spatially interpolated to the OMI pixel center (Boersma et al., 2007 and 2011). The eCF and CTP are obtained from the OMCLDO2 cloud product based on the $O_4$ absorption band at 477 nm assuming a Lambertian cloud with an albedo of 0.8 (Acarreta et al., 2004).

Two data sets of tropospheric $SO_2$ VCDs derived from OMI observations are validated in this study. One is the operational

level 2 OMSO2 planetary boundary layer (PBL) $SO_2$ data set (assuming $SO_2$ mostly in the PBL) provided via the NASA website (http://avdc.gsfc.nasa.gov). In the following this product is simply referred to as "OMI NASA". For the PBL $SO_2$ product the VCD is derived from the OMI-measured radiances between 310.5 and 340 nm using a principal component analysis (PCA) algorithm (Li et al., 2013). A fixed profile is used in the PBL retrievals for all OMI measurements (Krotkov et al., 2008). The second product is a data set extracted by a new OMI $SO_2$ retrieval algorithm developed by BIRA (Theys et

al., 2015). In the following this product is simply referred to as "OMI BIRA". It will form the basis of the algorithm for the operational level-2 $SO_2$ product to be derived from the upcoming TROPOspheric Monitoring Instrument (TROPOMI) instrument aboard the Sentinel-5 Precursor mission (Veefkind et al., 2012). $SO_2$ SCDs are retrieved in a window between 312 –326 nm using the DOAS technique and background corrected for possible bias. The $SO_2$ SCDs are converted to VCDs using a AMF look-up table, which is generated using the LInearized Discrete Ordinate Radiative Transfer (LIDORT) version

3.3 RTM (Spurr et al., 2001 and 2008). SFs for $SO_2$ are obtained from the IMAGES CTM (Müller and Brasseur, 1995) for individual measurements at a horizontal resolution of $2° \times 2.5°$ and at 40 vertical unevenly distributed levels extending from the surface to the lower stratosphere (44 hPa) (Stavrakou et al., 2013 and 2015). Like for the OMSO2 data set the cloud information is obtained from the OMCLDO2 cloud product.

The HCHO data set validated in this study is the OMI HCHO tropospheric VCD level 2 data retrieved by a DOAS algorithm

v14 developed at BIRA-IASB (De Smedt et al., 2015). This algorithm will also be applied to the upcoming TROPOMI instrument. HCHO SCDs are retrieved in the spectral window between 328.5–346 nm using the DOAS technique. After background corrections, HCHO SCDs are converted to tropospheric VCDs using AMFs from a look-up table generated by LIDORT with HCHO SFs obtained from the IMAGES CTM for individual measurements (Stavrakou et al., 2015). Also for this product the cloud information is obtained from the OMCLDO2 cloud product.

Here one important aspect should be noted: different AMF strategies are used in the DOMINO 2 $NO_2$ product and the BIRA $SO_2$ and HCHO products for eCF < 10%. For the $NO_2$ product the eCF and CTP are explicitly considered in the AMF simulations while for the $SO_2$ and HCHO products the clear sky AMFs are applied. These differences will be especially important for measurements in the presence of high aerosol loads (see section 3.6). For eCF>10%, a cloud correction based on the independent pixel approximation (IPA) (Cahalan et al., 1994) is applied for the three TG retrievals. It should also be





noted that observations of the outermost pixels (i.e. pixel numbers 1–5 and 56–60) and pixels affected by the so called "row anomaly" (see http://www.temis.nl/docs/omiwarning.html) were removed before the comparison.

## 2.3 NO$_2$, SO$_2$ and HCHO products derived from GOME-2

The GOME-2A and B instruments (Callies et al., 2000; Munro et al., 2006, 2016) are aboard the sun-synchronous
Meteorological Operational Satellite platforms MetOp-A and MetOp-B, respectively. MetOp-A (launched on 19 October 2006) and MetOp-B (launched on 17 September 2012) operate in parallel with the same equator crossing time of 09:30 LT. Before 15 July 2013 GOME-2A had the swath width of 1920km, corresponding to a ground pixel size of 80km×40km and a global coverage within 1.5 days. Since 15 July 2013, the GOME-2A swath width was changed to 960km with a ground pixel size of 40km×40km. The GOME-2A settings before 2013 are also applied to GOME-2B.

In this study, we validate the operational level 2 tropospheric NO$_2$ VCDs derived from the TM4NO2A version 2.3 product (Boersma et al., 2004) for GOME-2A and GOME-2B obtained from the TEMIS website. The NO$_2$ SCDs are retrieved in the 425-450 nm spectral window at BIRA with QDOAS (http://uv-vis.aeronomie.be/software/QDOAS/). The tropospheric NO$_2$ VCDs are obained from SCDs using the similar procedures as for the DOMINO 2 product. However, for the GOME-2 products the eCF and CTP are retrieved by the improved Fast Retrieval Scheme for Clouds from the Oxygen A-band
algorithm (FRESCO+) based on the measurements of the oxygen A-band around 760 nm (Wang et al., 2008) again assuming a Lambertian cloud.

Two SO$_2$ products derived from GOME-2A observations are included in the study. The first one is the operational level 2 O3M-SAF SO$_2$ product derived from GOME-2A observations (Rix et al., 2012 and Hassinen et al., 2016). In the following the product is simply referred to as "GOME-2A DLR". This product is provided via the EUMETSAT product navigator
(http://navigator.eumetsat.int) or the DLR EOWEB system (http://eoweb.dlr.de). The SO$_2$ SCDs are retrieved using the DOAS technique in the wavelength range between 315 and 326 nm. For the conversion of SCDs to VCDs, the AMFs are acquired from a AMF look-up table, which is generated using LIDORT 3.3. For the AMF computation, three types of SFs are assumed as Gaussian distributions with a FWHM of 1.5km around three central heights of 2.5km, 6km and 15km. Because for the SO$_2$ concentrations at Wuxi mostly anthropogenic pollutions is relevant, only the SO$_2$ product corresponding
to the central height of 2.5km is included in the validation study. The cloud information is obtained from GOME-2 measurements by the OCRA and ROCINN algorithms (Loyola et al., 2007) based on oxygen A-band observations at around 760 nm. The second product is provided from BIRA using the same retrieval algorithm as for the OMI BIRA SO$_2$ product, referred to as "GOME-2A BIRA". The same algorithm is also used to acquire the SO$_2$ data from GOME-2B observations. The product is referred to as "GOME-2B BIRA" in the following. The cloud properties used in the two products are derived
from GOME-2A/B observations using the FRESCO+ algorithm.





The HCHO tropospheric VCD level 2 products derived from GOME-2A and B observations (De Smedt et al., 2012 and 2015) are validated in this study. The same retrieval approach as for the OMI BIRA HCHO product is applied, but the cloud properties are derived from GOME-2A/B observations using the FRESCO+ algorithm.

## 3 Validation of the satellite data sets

In this section the daily and bi-monthly averaged $NO_2$, $SO_2$ and HCHO VCDs from OMI and GOME-2 are validated by comparisons with the tropospheric VCDs derived from the MAX-DOAS observations. Also the diurnal and weekly cycles from satellite observations are compared with those from the corresponding MAX-DOAS observations. Finally the influence of the SF and the effects of aerosols on the OMI products are discussed. The SFs from the CTM used for the OMI AMF calculations are compared to the SFs derived from MAX-DOAS.

### 3.1 Effects of variations of the coincidence criteria on the validation

Because of the large ground pixel size of the satellite observations, MAX-DOAS results are averaged over a time period around the satellite overpass time to (partly) compensate the effect of horizontal gradients of the TG concentrations. In principle the time period is a function of the satellite pixel size, the wind speed and the life time of the trace gases. Although some factors change frequently, here we use one fixed time period for the long-term comparisons for simplicity. In this study,
we test the effect on the satellite validation for four time periods including 1 hour, 2 hours, 3 hours and 4 hours around the satellite overpass time. Scatter plots of the average MAX-DOAS data over three time periods (1 hour, 3 hours and 4 hours) against those over 2 hours are shown in Fig. 3. The correlation coefficients are close to unity for all time periods. However, the slopes become systematically smaller for larger time periods (up to -10%) because of temporal smoothing. The results of the linear regressions and mean relative differences from the comparisons are also shown in Fig. 5a and will be discussed
below together with the effect of the selected coincidence area of the satellite products.

In principle for the satellite validation the satellite pixel closest to the MAX-DOAS instrument need to be selected. However, in order to minimise the random noise of the satellite data, it is useful to calculate the average of several satellite observations close to the measurement site (see e.g. Irie et al., 2012 and Ma et al., 2013). As selection criterion, a distance between the centre of the satellite pixel and the measurement site can be specified. This optimum distance depends on many
factors, such as the satellite ground pixel size, the selected time period over which the MAX-DOAS results are averaged, the expected horizontal gradients of the trace gas and the statistical uncertainty of the satellite data. A distance of < 20 km has been used for $NO_2$ comparisons (e.g. Ma et al., 2013 and Chan et al., 2015), 100 km for HCHO (De Smedt et al., 2015) and $SO_2$ (Theys et al., 2015). Irie et al. (2012) already found that the correlations and slopes of the linear regressions of the $NO_2$ tropospheric VCDs from OMI and GOME-2A against those from MAX-DOAS observations depend systematically on the
distance to the MAX-DOAS station.





We test the effect of the variation of the distance between 10 km to 75 km on the comparison between the satellite data (OMI and GOME-2) and the MAX-DOAS data for all three TGs. The areas for the four radii (10km, 20km, 50km and 75km) and the pixel sizes of OMI and GOME-2 are shown in the earth view image downloaded from the Google Earth service in Fig. 1 b-1. For distances larger than 20 km, the cities of Suzhou, Changzhou, Huzhou and Nantong are included in the area.

Because of transport of the pollutants between the cities and the different residence times, different horizontal distributions of the $NO_2$, $SO_2$ and HCHO VCDs are found around Wuxi as shown in Fig. 1 b-2, b-3 and b-4, respectively. HCHO has a smoother distribution than $SO_2$, which is smoother than $NO_2$. The satellite data for pixels with the distances of 0-10km, 10-20km, 20-50km and 50-75km to the MAX-DOAS station are compared with the MAX-DOAS results.

We compare both the results for individual satellite pixels and daily averages for the four radii with the average MAX-
DOAS data over 2 hours around the satellite overpass time. The comparisons for OMI $NO_2$, $SO_2$ and HCHO for pixels with distances of 0-10km, 10-20km, 20-50km and 50-75km are shown in Fig. 4a, b and c, respectively (the comparisons for pixels with the distances of <10km, <20km, <50km and <75km are shown in Fig. S1 in the Supplement). We use the $SO_2$ OMI product from BIRA for this study, because it shows in general a higher correlation with the MAX-DOAS data. We found that the linear regressions for the daily averaged data are quite similar to those for the individual pixel data. Only the
correlation coefficients are higher. The results of the linear regressions and the mean relative differences for the two distance categories as indicated in Fig. 4 and in Fig. S1 in the Supplement are shown in Fig. 5 b and c, respectively. The slopes decrease with increasing distance for the three gases. The decrease of the slopes (from 0.75 to 0.49 and $R^2$ from 0.66 to 0.29) are stronger for $NO_2$ than for $SO_2$ and HCHO. This finding is consistent with the typically stronger horizontal inhomogeneity of $NO_2$. The mean differences for HCHO show almost no dependence on the distance. This finding can be explained by the
more homogenous distribution of HCHO compared to $NO_2$ and $SO_2$. A significant decrease of the slopes from 0.73 to 0.50 and the $R^2$ from 0.65 to 0.44 is found for $NO_2$ with increasing distance over 20km. A decrease of the slope is also found for $SO_2$ for the distances larger than 20km. From these findings we conclude that 20km is a reasonable distance to select OMI $NO_2$ and $SO_2$ data for conditions similar to those at Wuxi. In contrast, for HCHO we select a distance of 50 km. Although for such distances the slope is smaller than for shorter distances, we find nearly identical mean differences. Because of this
finding and the rather high noise of the HCHO satellite data we select a distance of 50 km, for which the number of available measurements largely increases. The comparison of Fig. 5a and b indicates that the effect of time periods used for averaging the MAX-DOAS results on the validation study is much smaller than the effect of distances for selecting the satellite data. Thus we apply the time period of 2 hours around the satellite overpass time in this study.

Similar results for GOME-2 data as those for OMI shown in Fig. 5 are shown in Fig. 6. The O3M-SAF GOME-2A $SO_2$
product from DLR is used for this sensitivity study. Also for the GOME-2 $SO_2$ data set the effect of the horizontal coincidence criterion is larger than the effect of the time period for the averaging of the MAX-DOAS data is found. Thus also 2 hours around the satellite overpass time will be used for GOME-2 comparisons in this study. The largest changes of the slopes for the three trace gases are found around the distance of 10km, but the results for the selection criterion of 0-10km should be treated with care because of the low number of available measurements. The changes of the slopes for





distances larger than 20km are smaller than 0.06 for $NO_2$ and 0.04 for HCHO, but are larger for $SO_2$. However, the results of the linear regressions for $SO_2$ should again be treated with care because of the rather low correlation coefficients. From these results we select 50km as a reasonable distance for GOME-2 data of $NO_2$, $SO_2$ and HCHO.

In summary, in the following validation studies, the MAX-DOAS results are selected within the period from 12:30 LT to 14:30 LT for the comparisons with OMI and from 08:30 LT to 10:30 LT for the comparisons with GOME-2A/B. The OMI $NO_2$ and $SO_2$ (HCHO) data are selected for satellite pixels with the distance of <20km (<50km) from the Wuxi station. The GOME-2A/B data of the three species are selected for the distances < 50km.

### 3.2 Daily comparisons

The daily averaged satellite data for measurements within the chosen distances (see section 3.1) are compared with the daily averaged MAX-DOAS data within 2 hours around the satellite overpass time. To characterize the cloud effect on the comparisons, the comparisons are performed for different eCF bins of 0-10%, 10-20%, 20-30%, 30-40%, 40-50% and 50-100% for $NO_2$ and $SO_2$, and for eCF bins of 0-10%, 10-30%, 30-50% and 50-100% for HCHO.

### 1) $NO_2$

Figure 7a, b and c display scatter plots (and the parameters from the linear regressions) of the daily averaged $NO_2$ tropospheric VCDs derived from OMI, GOME-2A and GOME-2B products versus those derived from the corresponding MAX-DOAS measurements for eCF < 10%. Systematically higher correlation coefficients ($R^2$) for OMI than for GOME-2A/B are found. The systematic biases of the satellite data with respect to the MAX-DOAS data are quantified by the mean relative difference (MRD) calculated following Eq. 1:

$$\text{MRD} = \frac{\Sigma_1^n {(V_{s_i} - V_{M_i})}/{V_{M_i}}}{n} \qquad (1)$$

Here $V_{s_i}$ and $V_{M_i}$ represent the averaged TG VCDs from satellite observations and MAX-DOAS measurements on day i, respectively; n is the total number of the available days. The MRD is only 1% for OMI, and 27% and 30% for GOME-2A and GOME-2B, respectively.

The $R^2$, slopes and intercepts of the linear regressions, the MRD as well as the number of available days for the three satellite products are shown for the five eCF bins in Fig. 8. For OMI, $R^2$ decreases with increasing eCF; the slopes significantly change for eCF > 50% and the MRD drops to -40% for eCF > 40%. For GOME-2A, a steep decrease of $R^2$ for eCF > 30% is found. For GOME-2B a generally lower $R^2$ is found for eCF >30%.; the MRD indicates an increasing systematic overestimation for eCF > 30%. Thus we conclude that the cloud effect on OMI and GOME-2A/B $NO_2$ data becomes critical for eCF >40% and 30%, respectively.

### 2) $SO_2$:

Figure 9a, b, c, d and e display scatter plots of the daily averaged $SO_2$ tropospheric VCDs derived from the OMI NASA, OMI BIRA, GOME-2A DLR, GOME-2A and B BIRA products versus those derived from the corresponding MAX-DOAS measurements for eCF < 10%. $R^2$ and slopes are more close to unity for the OMI BIRA product than for the other products.



The MRDs indicate a similar systematic underestimation (-40% to -52%) by all products. There are fewer negative values in the OMI BIRA product than in other satellite products.

The $R^2$, slopes and intercepts of the linear regressions, the MRD as well as the number of the available days obtained for the five satellite $SO_2$ products are shown for the five eCF bins in Fig. 10. For the OMI BIRA product, a significant decrease of $R^2$ occurs for eCF > 10% together with a decrease of the slopes and the MRD. A steep increase of the MRD is found for eCF > 40%. Thus we conclude that the OMI BIRA $SO_2$ data are most accurate for eCF < 10%, while they might be still used for eCF of 10% to 40% with a 20% larger systematic negative bias than those for eCF < 10%. For the OMI NASA data, $R^2$, slope and MRD significantly decrease for eCF > 20%. $R^2$ for both GOME-2A data are low (<0.09) for all eCF bins, thus the linear regressions cannot yield meaningful information on the cloud effect. Almost constant MRDs are found for both GOME-2A $SO_2$ products for eCF<30%. For eCF>30% largely varying MRD are found, especially for the GOME-2A BIRA products. Thus we conclude that both GOME-2A products are most accurate for eCF < 30%. For the GOME-2B BIRA data, an obvious decrease of $R^2$ and slope is found for eCF > 10%, while for eCF>30% largely variable MRDs are found. Thus for the GOME-2B BIRA product we recommend to use observations with eCFs of <10%. $SO_2$ VCDs for eCF <30% might also be used, but are subject to larger uncertainties.

**3) HCHO:**

Because of the rather low atmospheric absorption of HCHO, the DOAS fit errors often dominate the total uncertainty of the HCHO satellite data (De Smedt et al., 2015). Thus systematic effects, e.g. caused by clouds, are more difficult to identify and quantify than for $NO_2$ and $SO_2$. Figure 11 shows the scatter plots of OMI HCHO VCDs versus those derived from MAX-DOAS observations for eCF < 30%. One important finding is that the $R^2$ for data with a fit error $< 7 \times 10^{15}$ molecules $cm^{-2}$ is better than the $R^2$ for all data. This indicates that the fit error dominates the random noise of satellite HCHO tropospheric VCDs. The mean fit error of the HCHO VCDs is $7 \times 10^{15}$ molecules $cm^{-2}$ for OMI data. Thus for further comparisons, we exclude the HCHO VCDs with fit error $> 7 \times 10^{15}$ molecules $cm^{-2}$ for OMI. However for the GOME-2A/B products, the filter for the fit error is not applied because in contrast to the OMI HCHO data we find a systematic dependence of the fit error on the retrieved HCHO tropospheric VCD (see Fig. S2 in the supplement).

If the additional filter of the fit error for the OMI product is applied, 48% of the total number of HCHO data is left for comparisons. In order to include sufficient numbers of data, we use broader eCF bins (0-10%, 10%-30%, 30%-50% and 50%-100%). Figure 12a, c and d display scatter plots of the satellite daily averaged data versus the MAX-DOAS data for eCF < 10% for OMI, GOME-2A and GOME-2B data, respectively. We found the best consistency for the GOME-2B product probably because of the weaker degradation of the instrument during the short time after launch. Nevertheless also other unknown reasons might play a role. One interesting finding is the better correlation of the OMI products for the eCF bin of 10% to 30% (see Fig. 12b) compared to the eCF < 10%. However, for eCF of 10% to 30% also a larger MRD of -34% (see Fig. 13) is found, which might be attributed to the special effect of clouds, namely the clear sky AMFs used in the retrievals for eCF<10% (see the last paragraph of section 2.2) .



The dependences of the results of the linear regression and the MRDs on the eCFs are shown in Fig. 13 for the three satellite instruments. For the OMI product a decrease of $R^2$ occurs for eCF > 30%, while for GOME-2A and GOME-2B, low $R^2$ are already found for eCF > 10%. Gradually increasing absolute values of the MRDs for all the satellite instruments are found for increasing eCF. We suggest that HCHO products for eCF < 30% should be used for the three satellite instruments

**3.3 Errors of Shape Factors from CTM and the effect on satellite VCD products**

The SF is an input for the calculation of satellite AMF, which is needed to convert the SCD to VCD (Palmer et al., 2001). Different retrieval algorithms acquire the SFs in different ways, mostly from a CTM for individual measurements or assuming a fixed SF (see section 2.2 and 2.3). The MAX-DOAS measurements acquire the vertical profiles of $NO_2$, $SO_2$ and HCHO from the ground up to the altitude of about 4km (depending on the measurement conditions), in which the

tropospheric amounts of the TGs is mostly concentrated. Thus the profiles derived from MAX-DOAS observations are valuable to evaluate the SFs used in the satellite retrievals and their effect on the AMFs and VCDs. Because the averaging kernels and SFs for individual satellite measurements are available only for the DOMINO $NO_2$, BIRA $SO_2$ and BIRA HCHO products derived from OMI observations, the three products are used to evaluate the effect of the SF in this section.

For the three selected products, the calculation of satellite tropospheric AMFs follows the same way introduced in Palmer et

al. (2001) as Eq. 2:

$$AMF = \int_{ground}^{tropopause} BAMF(z)SF(z)dz \tag{2}$$

Where BAMF(z) is the box AMF, which characterizes the measurement sensitivity as a function of altitude (z). The integration is done from the ground to the tropopause. The SFs of the TGs are obtained from different CTM (TM4 for $NO_2$, IMAGES for $SO_2$ and HCHO, see section 2.2). The profiles ($profile_M$) derived from MAX-DOAS can be converted to SF

($SF_M$) using Eq. 3:

$$SF_M(z) = \frac{profile_M(z)}{VCD_M} \tag{3}$$

where $VCD_M$ is the tropospheric VCD derived by an integration of the corresponding $profile_M$.

A similar relationship connects the BAMFs and averaging kernels (Eskes and Boersma, 2003):

$$AK(z) = \frac{BAMF(z)}{AMF} \tag{4}$$

The $SF_M$ can replace the SF from CTM ($SF_C$) to recalculate the AMF using Eq. 2. A similar study was recently conducted by Theys et al. (2015) and De Smedt et al. (2015) for OMI BIRA $SO_2$ and HCHO products over the Xianghe area. They demonstrated the improvements of the consistency between OMI VCDs and MAX-DOAS VCDs when using the $SF_M$ for the AMF calculation of the satellite products by 20%-50%. In our study we follow the same procedure.

**1)  $NO_2$**

The averaged $NO_2$ $SF_C$ for the measurements under clear sky with eCF < 10%, is compared to $SF_M$ in Fig. 14a. The differences between the averaged $SF_C$ and $SF_M$ shown in Fig. 14b indicate that the $NO_2$ $SF_C$ is considerably larger and



smaller than $SF_M$ in the layer below and above 0.4 km, respectively. The OMI VCDs ($VCD_{CTM}$) from the DOMINO $NO_2$ product based on $SF_C$ and the modified OMI VCDs ($VCD_{SM}$) based on $SF_M$ are plotted against the VCDs from MAX-DOAS observations in Fig. 14c. Very similar results for both $VCD_{CTM}$ and $VCD_{SM}$ are found. In Fig. 14e the relative differences of the AMFs using either $SF_C$ or $SF_M$ are shown. The differences are calculated in two ways: either the relative differences are first calculated for individual measurements, and then the individual relative differences are averaged. Alternatively first the AMFs of the individual measurements are averaged, and then the relative differences are calculated. The results in Fig. 14e show that for both calculations very similar results are obtained. The relative differences systematically increase with increasing eCF. For eCF<10% the relative differences are only 0.3%. The compensation of the negative and positive difference between $SF_C$ and $SF_M$ near the surface and at high altitudes contributes to the negligible SF effect on the AMF. The stronger effect of the SF on the AMF under cloudy sky conditions can be explained by the fact that the box-AMF below the cloud decrease strongly. This is the latitude range with the larges differences between $SF_C$ and $SF_M$. In general the agreement with MAX-DOAS VCDs by replacing $SF_C$ with $SF_M$ in the AMF calculation is only slightly improved for all the eCF bins. For large eCF, $VCD_{SM}$ is systematically larger than $VCD_{CTM}$ by 20% on average, consistent with AMF differences in Fig. 14e.

**2) $SO_2$**

The results shown in Figure 15a and b indicate that for eCF < 10%, the $SO_2$ $SF_C$ is considerably smaller and larger than $SF_M$ in the layer below and above 1 km, respectively. Since the BAMF increase with altitude (Fig. 14d) $SO_2$ $AMF_{CTM}$ are on average larger than $AMF_{MAX-DOAS}$ by 18% (Fig. 15e). In contrast to $NO_2$ the $VCD_{SM}$ agrees better with the MAX-DOAS VCDs than $VCD_{CTM}$, i.e. $R^2$ and slope increase from 0.47 to 0.60 and from 0.55 to 0.90, respectively (see Fig. 15c). Also the systematic bias of $VCD_{SM}$ is smaller than that of $VCD_{CTM}$, i.e. the MRD is -26% for $VCD_{SM}$ and -40% for $VCD_{CTM}$ (see black and red curves in Fig. 10).

For different eCF bins, the differences between $SO_2$ $SF_C$ and $SF_M$ (Fig. 15b) are slightly different from each other, and the BAMFs for large eCF are larger and smaller than those for low eCF at high and low altitudes, respectively. Also the relative differences between $AMF_{CTM}$ and $AMF_{MAX-DOAS}$ depend on eCF with larger differences for large eCF. However, the dependence of the differences on eCF is smaller than that for $NO_2$. Here it is interesting to note that the results shown in Figure 10 also showed a better consistency between the $SO_2$ $VCD_{SM}$ and the MAX-DOAS VCDs than for the $VCD_{CTM}$ for all the eCF bins.

**3) HCHO**

The results shown in Figure 16a and b indicate that for eCF < 10% the HCHO $SF_C$ is considerably smaller and larger than $SF_M$ in the layer below and above 1.7 km, respectively. Since the BAMF increases with altitude (Fig. 16d) HCHO $AMF_{CTM}$ is on average larger than $AMF_{MAX-DOAS}$ by 11% (Fig. 16e). Like for $SO_2$ the $VCD_{SM}$ agrees better with the MAX-DOAS VCD than $VCD_{CTM}$, i.e. $R^2$ and slope increase from 0.15 to 0.21 and from 0.44 to 0.61, respectively (see Fig. 16c). Also the systematic bias of $VCD_{SM}$ is smaller than that of $VCD_{CTM}$, i.e. the MRD is -10% for $VCD_{SM}$ and -18% for $VCD_{CTM}$ (see Fig. 13).



For different eCF bins, the dependences of differences between HCHO $SF_C$ and $SF_M$, and BAMFS on the eCFs are found similar to those of $SO_2$ (see Fig. 16b and d). Again, the large relative differences between $AMF_{CTM}$ and $AMF_{MAX-DOAS}$ are found for large eCF (see Fig. 16e). Figure 16e shows that for all the eCF bins the consistency between $VCD_{SM}$ and the MAX-DOAS VCD is better than for $VCD_{CTM}$.

4)  **Uncertainties of the SF from MAX-DOAS**

The previous study on Wuxi MAX-DOAS observations (Wang et al., 2016) demonstrated that the profile retrievals are not sensitive to altitudes above 1-2km, where the retrieved profiles are strongly constrained to the a-priori profiles. Thus the SFs at high altitudes could be underestimated by MAX-DOAS retrievals. This effect could be considerable especially for $SO_2$ and HCHO, because they typically extend to higher altitudes than $NO_2$ (Xue et al., 2010, Junkermann, 2009 and Wagner et
al., 2011). Because BAMFs of satellite observations are normally larger at high altitudes, the uncertainties of SFs from MAX-DOAS could cause an underestimation of $AMF_{MAX-DOAS}$, which further causes an overestimation of $VCD_{SM}$.

### 3.4 Comparisons of the bimonthly mean VCD

We calculate bi-monthly averaged tropospheric VCDs for eCF<30% for the coincident observations of the satellite instruments and MAX-DOAS (and also from the CTM simulations for the OMI products) from 2011 to 2014. The results for
$NO_2$, $SO_2$ and HCHO are shown in Fig. 17. The numbers of available days for each satellite products are also shown in the bottom panels of each subfigure.

1)  **$NO_2$**

For OMI good agreements with the MAX-DOAS VCDs are found both for the DOMINO and the improved VCDs using SFs from MAX-DOAS observations with slightly better agreement of the improved VCDs. GOME-2A and GOME-2B VCDs are
systematically larger than the MAX-DOAS VCDs by about $5 \times 10^{15}$ molecules cm$^{-2}$ on average. The overestimation could be attributed to the errors of the $NO_2$ SFs from TM4 (Pinardi et al., 2013). Systematic differences between the GOME2-A and GOME-2B VCDs are found, which can be partly explained by the different swath widths of both sensors after 15 July 2013. For the same reason also better agreement between GOME-2A and MAX-DOAS VCDs is found after summer 2013. The $NO_2$ VCDs simulated by TM4 for the OMI DOMINO product are much smaller than those from satellite and MAX-DOAS
observations. However the model data show a similar seasonality as the observational data.

2)  **$SO_2$**

For $SO_2$ large differences between the absolute values of the satellite and MAX-DOAS results are found, but all data sets show a similar seasonality with minima in summer and maxima in winter. The best agreement with MAX-DOAS results is found for the OMI BIRA $VCD_{SM}$, which displays an almost identical magnitude of the $SO_2$ annual variation (while still
showing a large bias). Interestingly, a much better agreement is found for the modified OMI $SO_2$ than for the OMI BIRA using the SF from the CTM. However the MAX-DOAS results are still significantly higher than the modified OMI products by about $10 \times 10^{15}$ molecules cm$^{-2}$ on average. Several reasons could contribute to the differences: 1) the horizontal gradient





of $SO_2$ (see Fig. 1) and the MAX-DOAS pointing direction to the North can contribute to the differences of about $3 \times 10^{15}$ molecules cm$^{-2}$. 2) The $SO_2$ cross section at 203K is applied in the current version of the OMI BIRA product. It was found that the temperature dependence of the $SO_2$ cross sections (Bogumil et al. 2003) should also be considered using e.g. a post-correction method (BIRA-IASB, 2016). The correction can increase $SO_2$ VCDs by up to $10 \times 10^{15}$ molecules cm$^{-2}$ with the

highest absolute changes in winter. 3) The surface albedo used in the retrieval of the OMI BIRA product is taken from the climatological monthly minimum Lambertian equivalent reflector (minLER) data from Kleipool et al. (2008) at 328 nm. We expect an uncertainty of the albedo of about 0.02. This will translate to an error of 15-20% of the $SO_2$ VCDs. 4) some unknown local emissions near the station might be underestimated by the satellites, but seen by the MAX-DOAS.

The BIRA GOME-2A/B and DLR GOME-2A data are well consistent with each other, but show large differences to the

corresponding MAX-DOAS results. The $SO_2$ VCDs simulated by IMAGES are systematically lower than the MAX-DOAS observations and show only a low amplitude of the seasonal variation.

### 3) HCHO

Relatively good agreements between the satellite and MAX-DOAS observations of HCHO are found for all data sets (except GOME-2A before summer 2013). For OMI a better agreement is found for the modified VCDs than for the original product,

with larger improvements of the OMI VCD in summer. GOME-2A/B products are consistent with each other but strongly underestimate the HCHO VCDs, especially in summer. It is interesting to note that the CTM results have a better consistency with the MAX-DOAS results than the OMI data. GOME-2A data before summer 2013 show the largest disagreement with the MAX-DOAS data. The reason for this finding is not clear, but might be related to the different swath width in that period.

**3.5 Diurnal variations characterized by combining the GOME-2A/B and OMI observations and the weekly cycle**

Because of the morning and afternoon overpass time of GOME-2 and OMI, respectively, several studies (e.g. Lin et al., 2010; De Smedt et al., 2015) investigated the differences of both data sets to characterize the diurnal variations of the TGs. In this section we perform a similar study, but include also MAX-DOAS data coincident to the satellite observations. We calculate the ratios between the bi-monthly mean tropospheric VCDs from GOME-2A/B and OMI (Ratio$_{Sat}$) for each species and the

corresponding ratios from the MAX-DOAS observations (Ratio$_{M-D}$). The results are shown in Fig. 18. The averaged Ratio$_{Sat}$ and Ratio$_{M-D}$ over the whole period are listed in Table 1. For $NO_2$, the Ratio$_{Sat}$ for both GOME-2 instruments show good agreement. Good agreement is also found for the seasonal variation with the MAX-DOAS results, but the absolute values differ. For $SO_2$, Ratio$_{M-D}$ and Ratio$_{Sat}$ are close to unity during a whole year, implying similar $SO_2$ VCDs around the overpass times of GOME-2 and OMI, but Ratio$_{Sat}$ shows also several positive and negative deviations from unity. For HCHO, in

general, good agreement between Ratio$_{Sat}$ and Ratio$_{M-D}$ for GOME-2A and GOME-2B is found (except some outliers of Ratio$_{Sat}$). Interestingly, both Ratio$_{Sat}$ and Ratio$_{M-D}$ are below unity indicating lower HCHO VCDs in the morning than in the afternoon.



We evaluate the weekly cycles of the VCDs of the TGs observed by satellite instruments and the corresponding MAX-DOAS. The weekly cycles are shown in the Fig. S3 in the supplement. In general only both GOME-2 instruments and corresponding MAX-DOAS measurements observed considerable weekly cycles for $NO_2$.

**3.6 Aerosol effects on the satellite results**

In this section the aerosols effects on the satellite products are investigated. For that purpose we focus on the OMI products (the OMI BIRA product for $SO_2$) because of their marked consistency with MAX-DOAS results. We selected satellite observations for eCF<10%, for which a potential cloud contamination is small. Moreover, especially over polluted regions like Wuxi, eCF larger than zero often indicates the effect of aerosols rather than that of clouds. In Fig. 19a-1, b-1 and c-1 the differences of the TG VCDs between OMI and MAX-DOAS observations for individual OMI pixels are plotted against the aerosol optical depths (AODs) derived from the MAX-DOAS observations (Wang et al. 2016). We find the increasing negative bias with increasing AOD indicating the effect of aerosols on the satellite retrievals. Moreover, for $NO_2$ we find that the strongest negative biases are obtained for large CTP, indicating the presence of aerosols rather than of clouds. To skip measurements which are probably affected by remaining clouds, in Fig. 19 a-2, b-2 and c-2, we only show data for eCF <10% and CTP>900 hPa. We find that the stronger negative biases are generally related to a larger eCF, especially for $NO_2$. In summary we conclude that the OMI TG VCDs tends to underestimate the true TG VCDs with increasing AOD. Here it is important to note that in contrast to the DOMINO $NO_2$ product, a clear-sky AMF is applied in the retrieval of the OMI BIRA $SO_2$ and HCHO products for eCF<10%. For the DOMINO $NO_2$ product, the AMF is calculated assuming a Lambertian cloud using the simultaneously derived eCF and CTP (see below). Since aerosols affect these cloud products, this correction is often referred to as 'implicit aerosol correction' (Chimot et al., 2016).

To further characterize the influence of applying either the clear sky AMF or the implicit aerosol correction in the following AMFs based on typical conditions of aerosols and trace gases are calculated. In a first step we characterize the typical aerosol-induced eCF and CTP over the Wuxi station.

For that purpose we select six clear days with substantial aerosol pollution. We checked that the selected days were indeed cloud-free based on RGB images from the MODIS instrument operated on the Aqua satellite with an overpass time eight minutes later than OMI. The MODIS images are obtained from the MODIS Rapid Response website, NASA/GSFC (http://aeronet.gsfc.nasa.gov/cgi-bin/bamgomas_interactive) (Kaufman, 2002). In addition to the MODIS images we also checked time series of the AOD from MAX-DOAS and the nearby Taihu AERONET station (Holben et al. 1998, 2001). The MODIS images and time series of the AODs are shown in the Fig. S4 in the Supplement. In Table 2, the daily averaged AODs derived from MAX-DOAS observations, and the eCF and CTP derived from OMI observations are shown for the six days. The aerosol-induced eCF and CTP range from 4% to 9% and from 830 to 995 hPa, respectively. The averaged vertical aerosol extinction profiles for cloud-free sky conditions are shown in Fig. S5.

For one typical nadir satellite observation geometry (40 ° SZA, 180 ° RAA and 30 ° VZA), we simulated BAMFs for $NO_2$ at 435nm, HCHO at 337 nm and $SO_2$ at 319 nm using the RTM SCIATRAN 2.2 (Rozanov et al., 2005). The simulations are





performed for four scenarios: 1) pure Rayleigh scattering conditions ($BAMF_{clear-sky}$); 2) including the explicit MAX-DOAS aerosol profiles ($BAMF_{explicit}$); 3) including aerosol-induced eCF of 5% and CTP of 1000hPa (near the surface) ($BAMF_{low-cloud}$); 4) including aerosol-induced eCF of 5% and CTP of 900hPa (cloud height of about 1km) ($BAMF_{high-cloud}$). The latter two cases represent the implicit aerosol correction and the same cloud model (Lambertian surface with an albedo of 0.8) as in the OMI products is used. The surface albedo is set to 0.1.

In Fig. 20a the resulting BAMFs for the different TGs and aerosol and cloud assumptions are shown. For all TGs similar results are obtained: compared to the clear sky the BAMFs for the explicit aerosol simulations are decreased close to the surface and increased for higher altitudes. For the 'low cloud scenario' largely increased BAMFs are found for all altitudes. For the 'high cloud scenario' largely increased BAMFs are found for high altitudes, whereas the BAMFs close to the surface are similar or slightly lower than the clear sky BAMFs. The differences of the BAMFs compared to the clear sky BAMFs are shown in Fig. 20 b and c.

Finally, we calculate AMFs of $NO_2$, $SO_2$ and HCHO for the four simulated BAMFs using typical SFs (shown in Fig. S5) derived from MAX-DOAS results or CTM simulations by Eq. 2. The derived AMFs are shown in Fig. 21. For most cases the best agreement with the AMFs derived for the explicit aerosol profiles is found for the clear sky AMFs. In contrast, assuming an implicit aerosol correction can lead to large deviations, especially for the low cloud scenario. Overall similar results are found for the SF derived from MAX-DOAS or CTM.

These findings are consistent with the aerosol effects on the OMI DOMINO $NO_2$ data shown in Fig. 19. In summary we conclude that for aerosol loads like those over Wuxi the implicit aerosol correction typically causes larger bias of the satellite TG VCDs than the clear-sky assumption. Thus if no explicit aerosol information is available, we recommend to apply the clear-sky AMFs for eCF<10%, especially for CTP>900hPa.

## 4 Conclusions

Tropospheric VCDs of $NO_2$, $SO_2$, HCHO derived from OMI, GOME-2A/B observations are validated using MAX-DOAS measurements in Wuxi, China from May 2011 to Dec 2014. The tropospheric VCDs and vertical profiles of aerosols and trace gases derived from the Wuxi MAX-DOAS observations using the PriAM OE-based algorithm are applied in this validation study.

Before the data sets are compared in a systematic way, the effects of the spatial and temporal coincidence criteria for the MAX-DOAS results and the satellite data are evaluated in detail. We find that the temporal scale over which the MAX-DOAS data are averaged has only a small effect on the comparison results. In contrast, the spatial scale over which the satellite data are averaged has a strong effect for the three species. However, a smaller effect is found for HCHO than for $NO_2$ and $SO_2$, which is explained by the weaker horizontal gradient of the HCHO distribution. Based on our results we recommend using OMI products within distances to the MAX-DOAS station of 20km for $NO_2$ and $SO_2$, and 50km for



HCHO. For GOME-2A/B, which has a larger ground pixel size, we recommend to use data within distances of 50km for $NO_2$, $SO_2$ and HCHO.

We compare the daily averaged tropospheric VCDs from satellite products with the corresponding MAX-DOAS results under clear sky conditions (eCF<10%). For $NO_2$: good agreement ($R^2$ of 0.73 and systematic bias of 1%) is found for the DOMINO product. For both GOME-2 products (TM4NO2A) much weaker correlation ($r^2$ of 0.33 for GOME-2A and 0.2 for GOME-2B) is found with the same systematic bias of about 30%. For $SO_2$: the OMI BIRA product has a much better correlation coefficient ($R^2$ of 0.47) than the OMI NASA product ($r^2 = 0.12$), the GOME-2A BIRA product ($r^2 = 0.07$), the GOME-2A DLR product ($R^2 = 0.09$) and the GOME-2B BIRA product ($r^2 = 0.28$). All of these products systematically underestimate the $SO_2$ tropospheric VCDs by about 40% to 57%. For HCHO: the best agreement is found for the GOME-2B product with $R^2$ of 0.53 and a systematic bias of -12%. The OMI and GOME-2A products have lower $R^2$ of 0.17 and 0.18 with the same systematic bias of about -20%, respectively.

In general, we expect that the VCDs from MAX-DOAS observations have much lower uncertainties than those from satellite observations. However we should also consider the total uncertainties of the MAX-DOAS VCDs of $NO_2$, $SO_2$ and HCHO of about 25%, 31% and 54%, respectively (Wang et al., 2016). Moreover, MAX-DOAS has low sensitivity to high altitudes, normally above 1-2km. This can cause an underestimation of the VCDs retrieved from MAX-DOAS. The effect depends on the vertical distribution of the species, the atmospheric visibility, and the observation geometry of the MAX-DOAS instrument. In this study we do not discuss these issues in more detail. This should be done in further studies. Nevertheless, the sensitivity of MAX-DOAS observations to the boundary layer is much larger than of satellite observations. This is the altitude range in which the pollutants are usually accumulated. Thus it is reasonable to assume that the systematic differences between both data sets are mainly attributed to the errors of satellite observations.

The cloud effects on the MAX-DOAS results and satellite products are discussed. Under partial cloud coverage, the cloud effects on the MAX-DOAS results are negligible. The consistency (correlations and systematic bias) of satellite data with MAX-DOAS results deteriorates with increasing eCF. The cloud effects become significant for eCF > 40% for the OMI DOMINO $NO_2$ product, >30% for the GOME-2A\B $NO_2$ products, > 10% for the OMI BIRA $SO_2$ product, >20% for the OMI NASA $SO_2$ product, >30% for the GOME-2A/B BIRA $SO_2$ products and >30% for all HCHO products. Note that the conclusions are for the original satellite products, namely using SF from CTM or assumed fixed SF. In addition, the different thresholds of eCF could also be related to the properties of the different cloud products. This effect is not discussed in this paper, and is valuable to be further studied.

In the OMI DOMINO $NO_2$, OMI BIRA $SO_2$ and HCHO products, the a-priori SFs of the trace gases are obtained from CTM. We compare these SFs with those derived from MAX-DOAS observation and find substantial differences. We investigate the effect of using the MAX-DOAS SFs in the satellite retrievals. Under clear sky conditions, including the SFs from MAX-DOAS changes the $SO_2$ and HCHO AMFs by about 18% and 11%, respectively, but has almost no impact on the $NO_2$ AMFs. We find that the modified satellite VCDs show much better agreement with the MAX-DOAS results (showing considerably higher correlation coefficients $R^2$ and smaller systematic biases) than the original satellite data. The improvement is the





strongest for periods with large trace gas VCDs, namely $NO_2$ and $SO_2$ in winter and HCHO in summer. In the period, $NO_2$, $SO_2$ and HCHO VCD change by up to 10%, 47% and 35%, respectively. We also found that using the MAX-DOAS SFs in the satellite retrievals has the strongest effect for increasing eCF. This finding is mainly caused by the shielding effect of clouds on the satellite observations. In addition the low sensitivity of MAX-DOAS could underestimate the SFs of the trace gases at high altitude (above 1-2km), especially for $SO_2$ and HCHO. This effect could cause the underestimation of AMFs and overestimation of VCDs by using the MAX-DOAS SFs.

The relative seasonal variations of the bi-monthly mean $NO_2$, $SO_2$, HCHO tropospheric VCDs from the different satellite products agree well with the corresponding MAX-DOAS results. The best consistency is found for the OMI DOMINO $NO_2$ product. A systematic overestimation of the $NO_2$ VCDs is found for GOME-2A\B $NO_2$ products. All $SO_2$ satellite products show similar $SO_2$ VCDs and a systematic underestimation of about $20 \times 10^{15}$ molecules cm$^{-2}$. Based on the studies on the OMI BIRA product, the systematic underestimation could be attributed to a combined effect of errors of SFs, horizontal gradients of the $SO_2$ distribution, the temperature dependence of the $SO_2$ cross section, and uncertainties from the surface albedo and local emissions. The OMI NASA and the GOME-2A BIRA and NASA $SO_2$ products show a larger random variability than the OMI and GOME-2B BIRA $SO_2$ products. All OMI and GOME-2A/B products systematically underestimate the tropospheric HCHO VCDs by about $5 \times 10^{15}$ molecules cm$^{-2}$, while showing a similar seasonality as the MAX-DOAS results.

We compared the diurnal variations by combining GOME-2A/B (morning overpass) with OMI (afternoon overpass) observations with the corresponding MAX-DOAS observations. For $NO_2$ higher values are found in the morning, while for HCHO higher values are found in the afternoon. For $SO_2$ no significant diurnal cycle was found. For the MAX-DOAS data similar results were obtained, but the $NO_2$ satellite products systematically overestimate the magnitude of $NO_2$ diurnal variation compared to the MAX-DOAS data. No significant weekly cycle was found for the three trace gases in the satellite and MAX-DOAS data.

Finally we studied the aerosol effect on the OMI products. We found an increasing underestimation of OMI $NO_2$, $SO_2$ and HCHO products with increasing AOD by up to 8%, 12% and 2%, respectively.. The aerosol effects on the different satellite products are different, because different strategies for the calculation of AMFs are used: for the OMI DOMINO $NO_2$ product an implicit aerosol correction is applied based on the OMI cloud products. In contrast, for the BIRA $SO_2$ / HCHO products AMFs for clear-sky are used for eCF<10%. We investigated the aerosol effect on the cloud products (eCF and CTP) on six cloud-free days with pure aerosol pollutions. Aerosol-induced eCF and CTP between 4% and 9% and between 830 and 995 hPa are found, respectively. Our results indicate that the implicit correction could cause a strong underestimation of tropospheric VCDs by up to about 45%, 77% and 100% for $NO_2$, $SO_2$ and HCHO, respectively. For conditions with eCF <10% and CTP>900 hPa the AMFs based on the cloud products can lead even to larger errors than the AMFs based on the clear-sky assumption. Thus it is recommended to apply the clear-sky AMFs in such cases if explicit aerosol information is not available.



***Acknowledgements:*** We thank Wuxi CAS Photonics Co. Ltd for their contributions in operating the MAX-DOAS instrument. We thank the Institute of Remote Sensing / Institute of Environmental Physics, University of Bremen, Bremen, Germany for their freely accessible RTM SCIATRAN program. We thank Goddard Space Flight Center, NASA for their freely accessible archive of AERONET and MODIS data. We thank Prof. Ma Ronghua in Nanjing Institute of Geography & Limnology Chinese Academy of Sciences for his effort to operate the Taihu AERONET station. We thank the Royal Netherlands Meteorological Institute for their freely accessible archive of OMI tropospheric $NO_2$ data. This work was supported by Max Planck Society-Chinese Academy of Sciences Joint Doctoral Promotion Programme, and Monitoring and Assessment of Regional air quality in China using space Observations, Project Of Long-term sino-european co-Operation (MarcoPolo), FP7 (Grant No: 606953) and National Natural Science Foundation of China (Grant No.: 41275038).

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





## Figures & Tables



Figure 1: Wuxi city, in which the MAX-DOAS instrument is operated, is marked by the red dot in subfigure (a-1). Subfigures (a-2), (a-3) and (a-4) show maps of the averaged tropospheric VCDs of NO$_2$ from DOMINO 2, SO$_2$ and HCHO from BIRA derived from OMI observations over eastern China in the period from 2011 to 2014, respectively. The black dots indicate the location of Wuxi. Subfigure (b-1) shows the earth image around Wuxi MAX-DOAS station from google earth service; the rectangles indicate the ground pixel sizes of the different satellite





instruments used in this study. (GOME2-A phase 1 and phase 2 corresponding to the periods before and after 15 July 2013); the circles indicate areas with different radii around Wuxi. The subfigures of (b-2), (b-3) and (b-4) show averaged VCDs of $NO_2$, $SO_2$ and HCHO for the same area as shown in (b-1); the black dots indicate the location of Wuxi and the green circles have a radius of 75km.

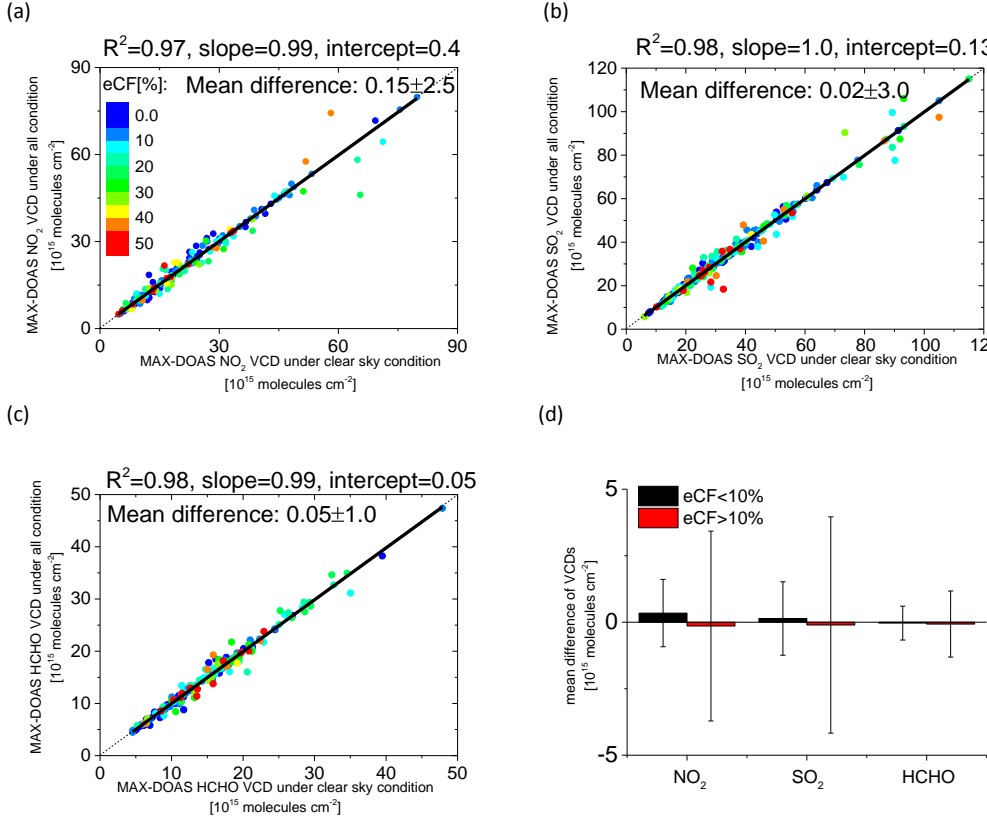

**Figure 2: Daily averaged (during two hours around the OMI overpass time) $NO_2$ (a), $SO_2$ (b) and HCHO (c) tropospheric VCDs derived from MAX-DOAS observations under all sky conditions plotted against those under clear sky conditions. The colours indicate the eCF. The correlation coefficients, slopes, intercepts and mean differences ± standard deviation are displayed in each subfigure. The mean differences for eCF <10% and >10% are plotted in subfigure (d) with the error bars denoting the respective standard deviations.**



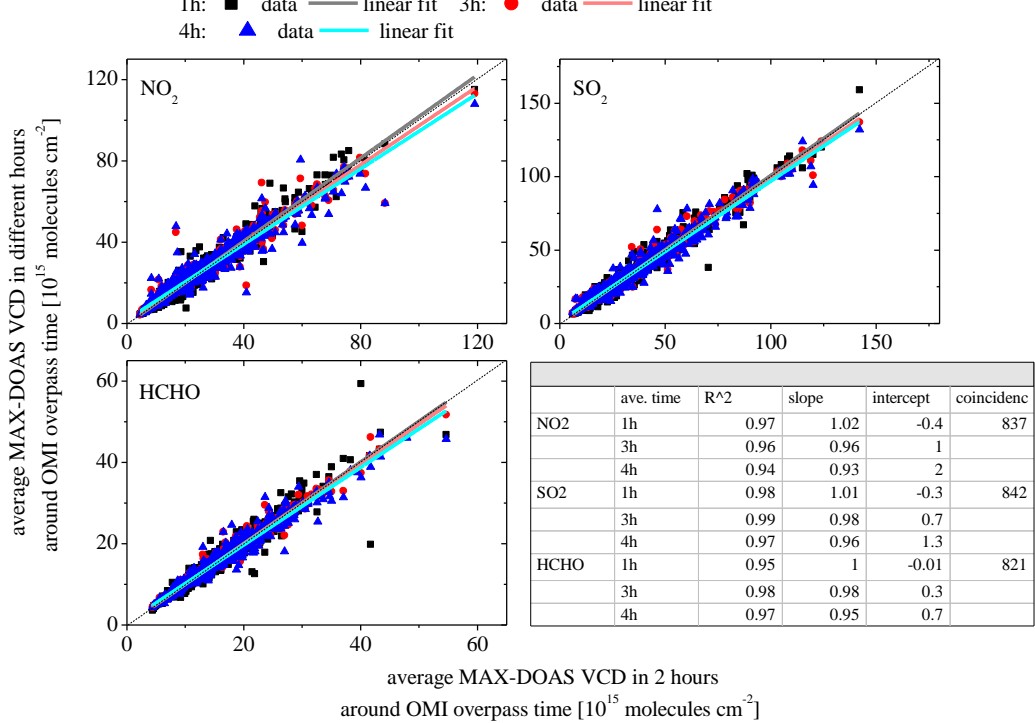

| | ave. time | R^2 | slope | intercept | coincidenc |
|---|---|---|---|---|---|
| NO2 | 1h | 0.97 | 1.02 | -0.4 | 837 |
| | 3h | 0.96 | 0.96 | 1 | |
| | 4h | 0.94 | 0.93 | 2 | |
| SO2 | 1h | 0.98 | 1.01 | -0.3 | 842 |
| | 3h | 0.99 | 0.98 | 0.7 | |
| | 4h | 0.97 | 0.96 | 1.3 | |
| HCHO | 1h | 0.95 | 1 | -0.01 | 821 |
| | 3h | 0.98 | 0.98 | 0.3 | |
| | 4h | 0.97 | 0.95 | 0.7 | |

**Figure 3: Averaged NO₂ (a), SO₂ (b) and HCHO (c) tropospheric VCDs derived from MAX-DOAS observations in time periods of 1 hour (black dots), 3 hours (red dots) and 4 hours (blue dots) around the OMI overpass time plotted against those in the time period of 2 hours around the OMI overpass time. The linear regression lines for each time period and each species are plotted in each subfigure. The corresponding parameters are listed in the table.**

(a)

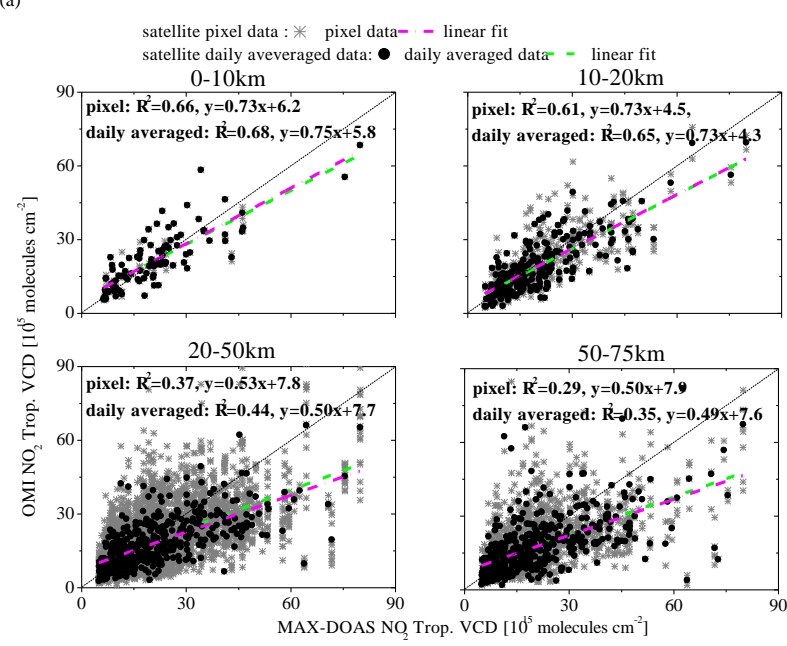

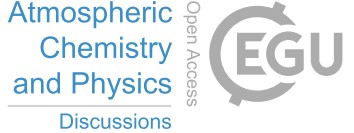

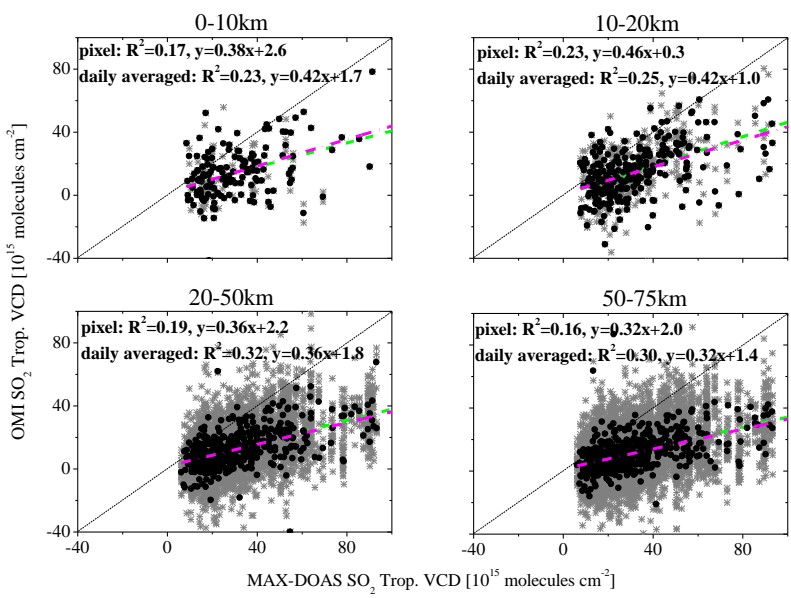

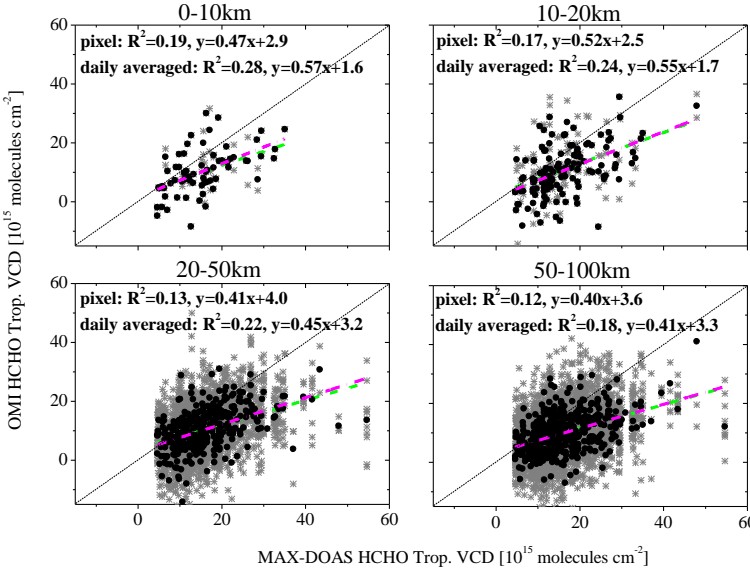

**Figure 4: Tropospheric VCDs of NO₂ (a), SO₂ (b) and HCHO (c) derived from OMI observations for pixels within the distance bins of 0-10km, 10-20km, 20-50km and 50-75km away from the Wuxi MAX-DOAS station plotted against the coincident MAX-DOAS results. Only OMI data for the eCF<30% are included. For HCHO, only the data for a fit error < 7×10¹⁵ molecules cm⁻² are included. The grey crosses and black dots show the data for individual satellite pixel and daily averaged data (averaged during two hours around the OMI overpass time), respectively. The linear regression lines and the parameters are shown in each subfigure for the pixel data (green dash lines) and daily averaged data (magenta dash-dot lines), respectively.**





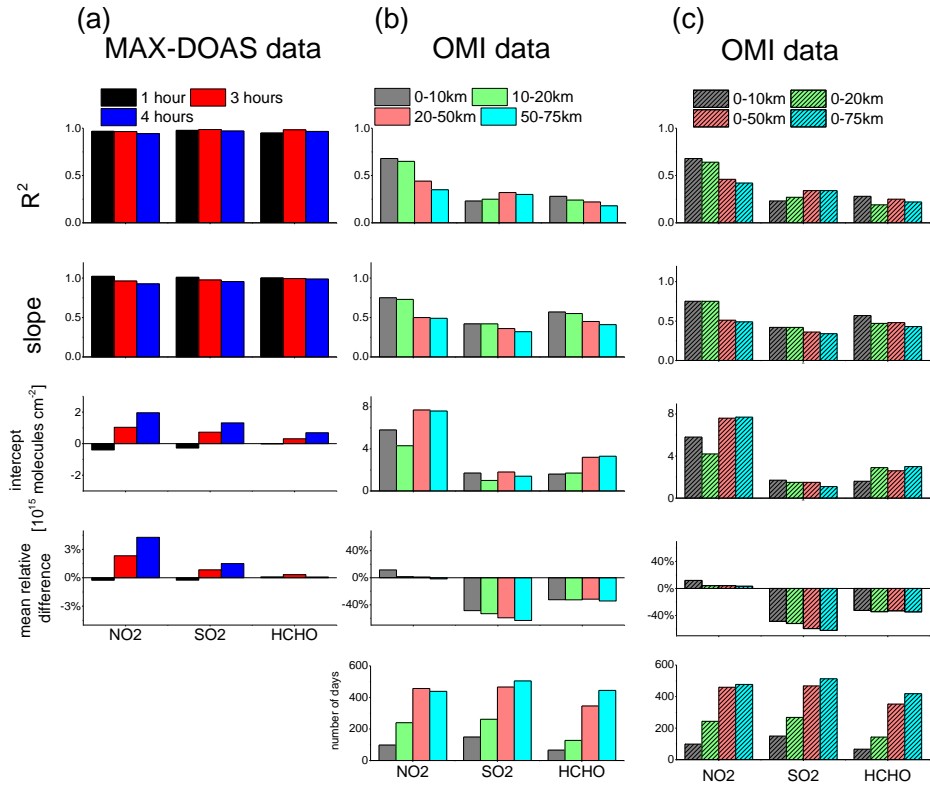

**Figure 5: (a) $R^2$, slope and intercept of the linear regressions as well as the mean relative differences of the averaged MAX-DOAS tropospheric VCDs of NO$_2$, SO$_2$ and HCHO in the time periods of 1 hour, 3 hours and 4 hours around the OMI overpass time compared to those in the time period of 2 hours. (b) $R^2$, slope and intercept of the linear regressions as well as the mean relative differences of the averaged OMI tropospheric VCDs of NO$_2$, SO$_2$ and HCHO for the pixels within the distance bins of 0-10km, 10-20km, 20-50km and 50-75km compared to the coincident MAX-DOAS results. At the bottom also the numbers of the days for each comparison are shown. (c) Similar with (b), but for distance bins of 0-10km, 0-20km, 0-50km and 0-75km.**





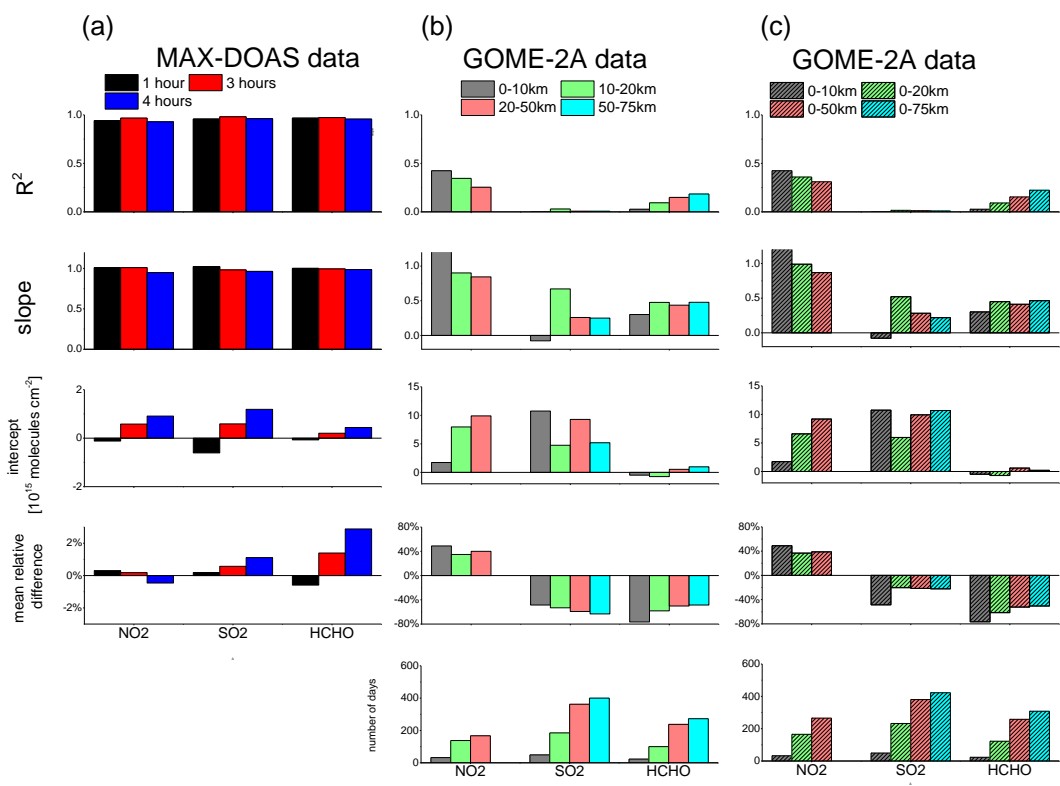

Figure 6: same as Fig. 5, but for GOME-2A data.

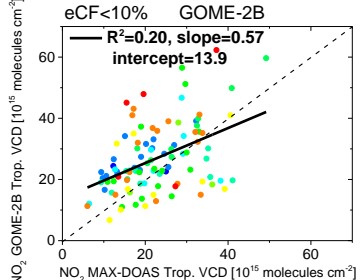

Figure 7: Daily average NO$_2$ tropospheric VCDs derived from OMI (a), GOME-2A (b) and GOME-2B (c) compared with the corresponding



**MAX-DOAS data for eCF<10%. The colors indicate the eCF.**

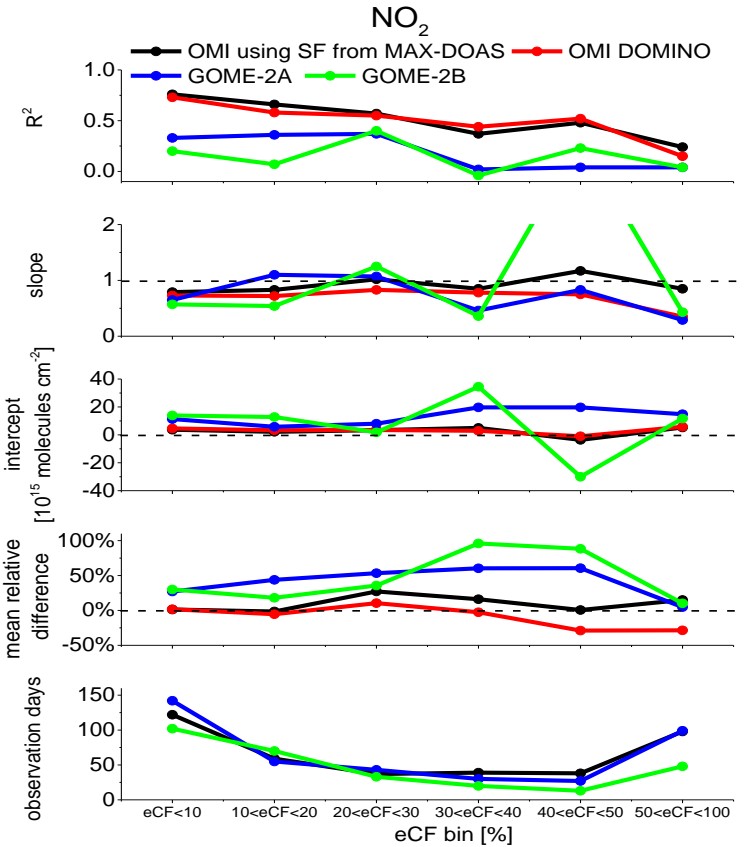

**Figure 8: $R^2$, slopes, intercepts, mean relative differences (and the number of available days) derived from the comparisons of the $NO_2$ VCDs from different satellite instruments to the MAX-DOAS results for the different eCF bins. Note that the black and red curves represent the improved OMI VCDs with the a-priori shape factors derived from Wuxi MAX-DOAS observations (see section 3.3) and for the DOMINO 2 product, respectively.**

(a)                                                    (b)

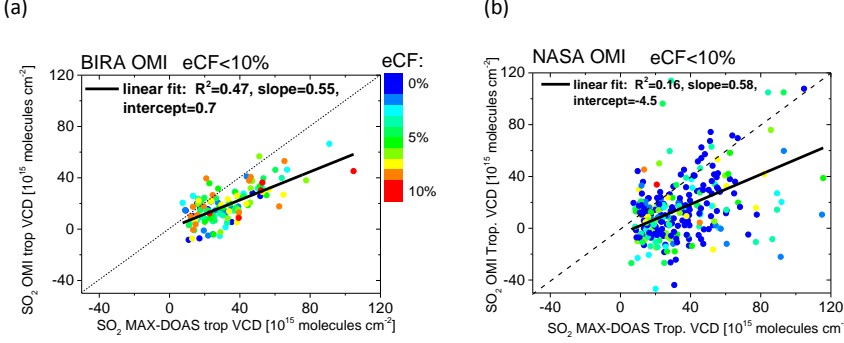



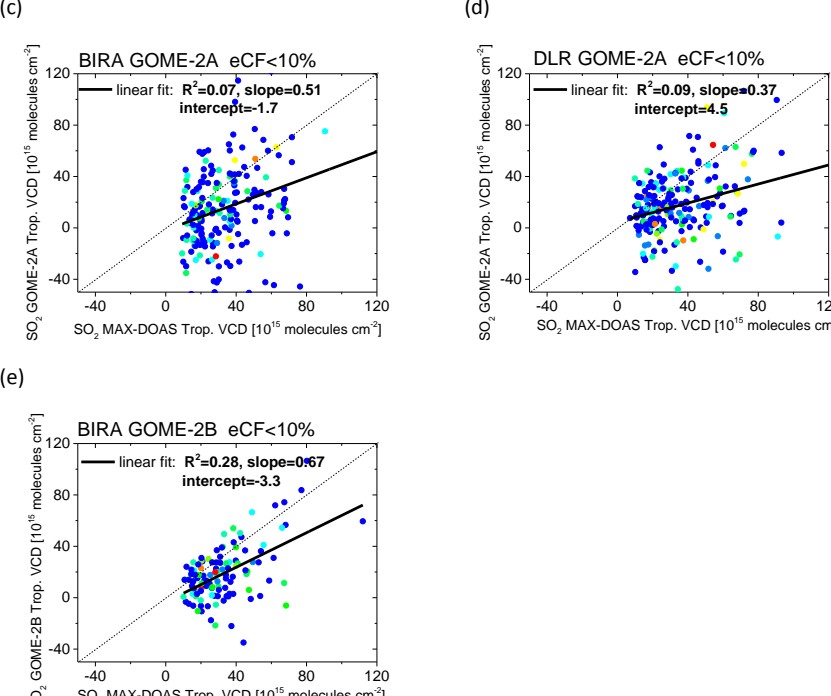

**Figure 9: Daily averaged OMI SO₂ tropospheric VCDs from BIRA (a) and NASA (b), GOME -2A SO₂ tropospheric VCDs from BIRA (c) and DLR (d) and GOME-2B SO₂ tropospheric VCDs from BIRA (e) for eCF < 10% plotted versus the coincident MAX-DOAS results. The colors indicate the eCF.**







**Figure 10: Same as figure 8 but for SO₂.**



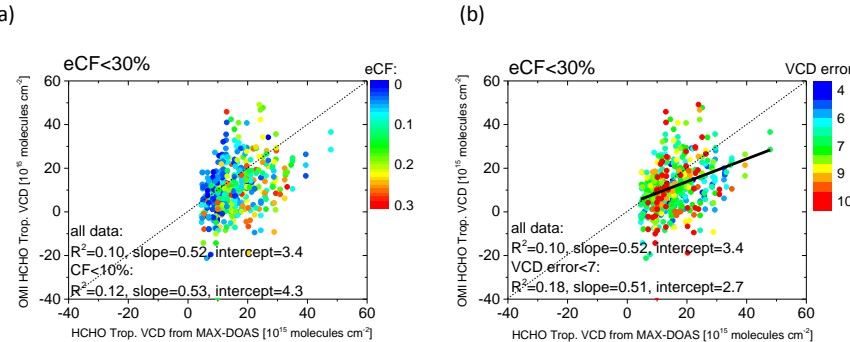

**Figure 11: (a) HCHO tropospheric VCDs for OMI pixels for eCF<30% are plotted against those derived from MAX-DOAS observations with the color map of eCF; the linear regression parameters are acquired for eCF<30% and for eCF<10%, respectively. (b) Scattered plots are same as in (a), but with the color map of VCD fit error; linear regression parameters are acquired for all data and for VCD fit error $<7\times10^{15}$ molecules cm$^{-2}$.**

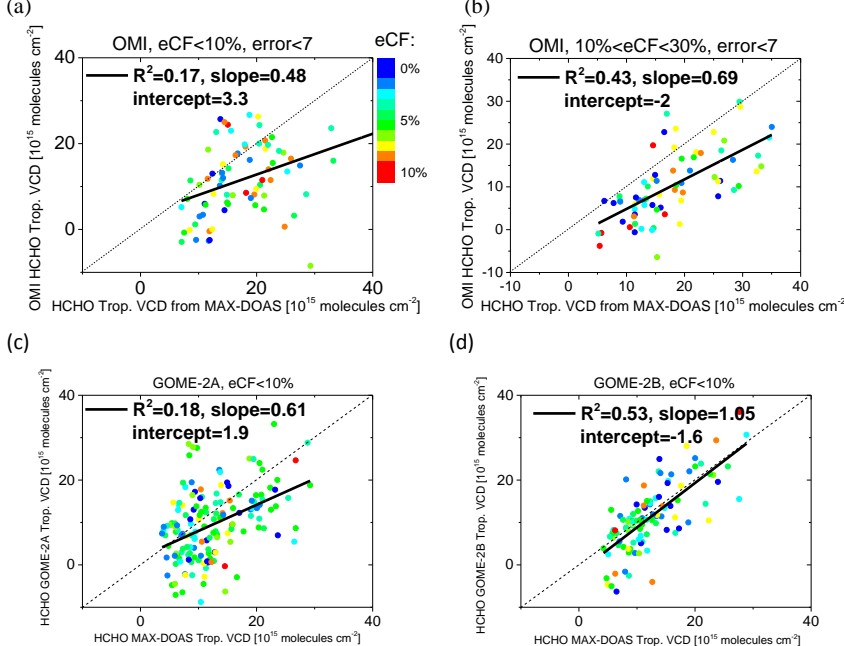

**Figure 12: Same as Fig. 7 but for HCHO.**





Figure 13: Same as Fig. 8 but for HCHO.





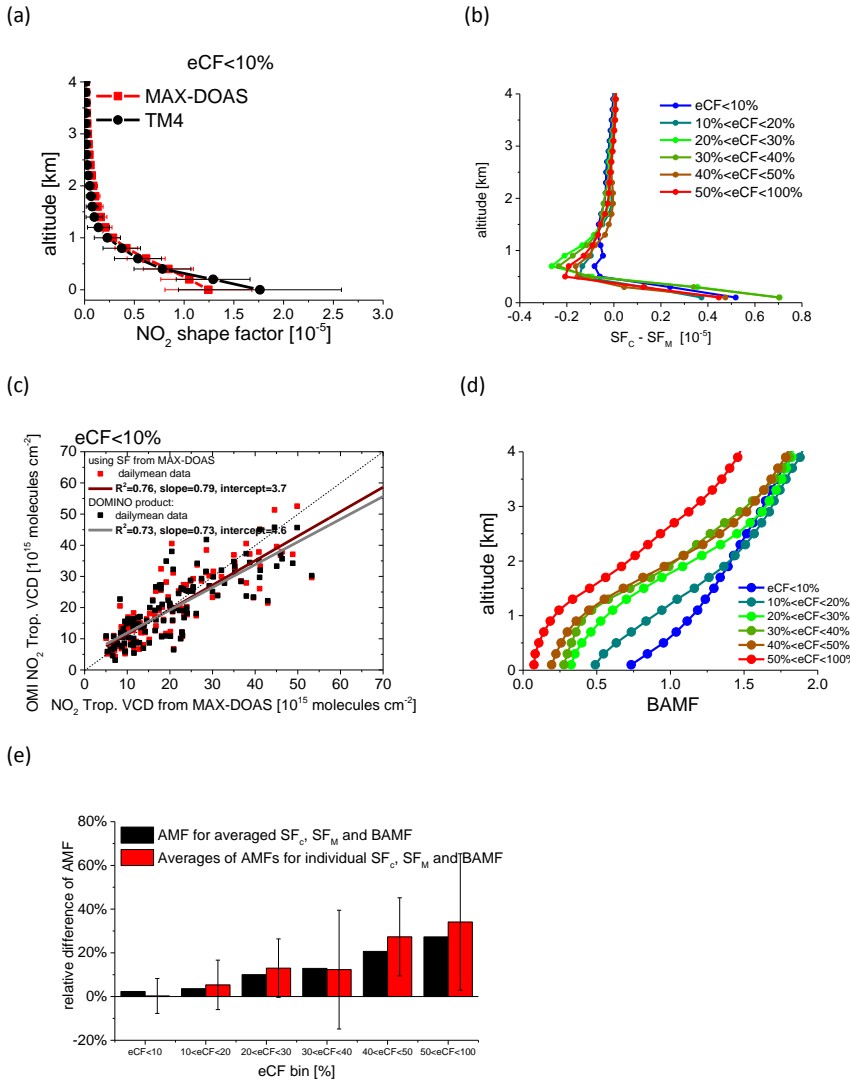

**Figure 14: (a)** Average $NO_2$ SFs and standard deviations derived from the MAX-DOAS observations and from the TM4 CTM (for the DOMINO product) for eCF<10%. **(b)** Averaged difference between the $NO_2$ SF from CTM ($SF_C$) and from MAX-DOAS ($SF_M$) for different eCF bins. **(c)** Daily averages of the original DOMINO $NO_2$ product and modified $NO_2$ product (based on MAX-DOAS SF) plotted against those from MAX-DOAS for eCF<10%. **(d)** Averaged BAMF for satellite observation for different eCF bins. **(e)** Relative difference (RD) of satellite AMF using $SF_C$ ($AMF_{CTM}$) or $SF_M$ ($AMF_{MAX-DOAS}$) for different eCF bins. The error bars indicate the standard deviation of the RDs for each eCF bin. Black columns denote the RDs derived from the averaged $SF_C$, $SF_M$ and BAMF (shown in subfigure (b) and (d)); red columns denote the averaged RDs for individual $SF_C$, $SF_M$ and BAMF of each satellite observation.





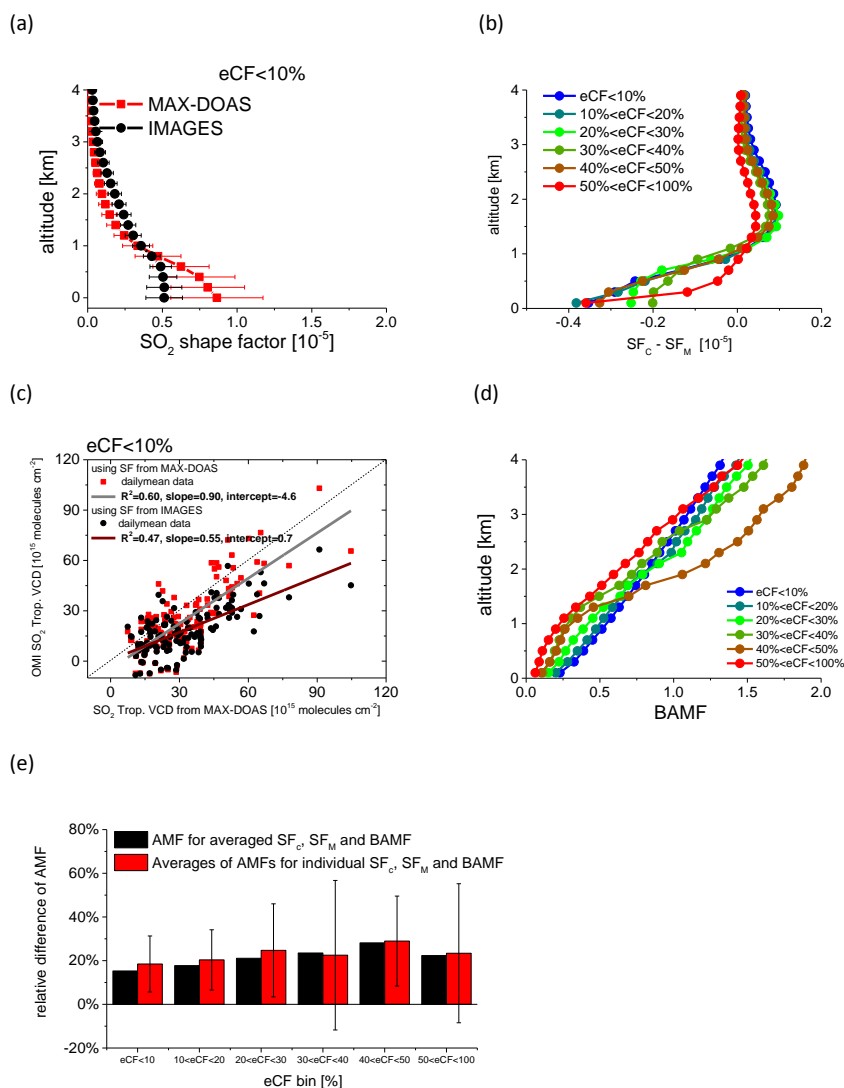

**Figure 15: Similar as Fig. 14 but for the OMI BIRA SO$_2$ product. Note that the SF for the OMI BIRA product is obtained from the IMAGES CTM.**

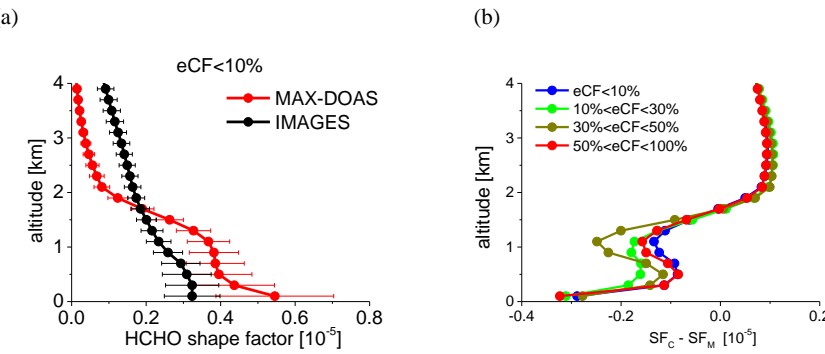



(c)

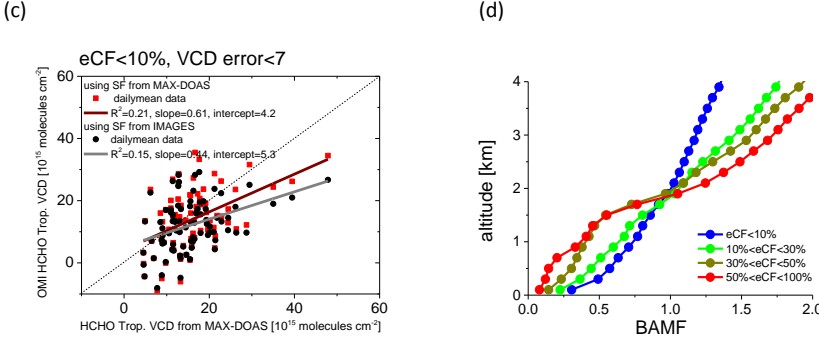

(d)

(e)

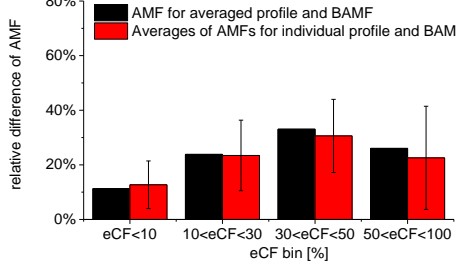

**Figure 16: Same as Fig. 14 but for the OMI BIRA HCHO product and eCF bins of 0-10%, 10%-30%, 30% -50% and 50% -100%. Note that the SF for the OMI BIRA product is obtained from the IMAGES CTM.**

(a)

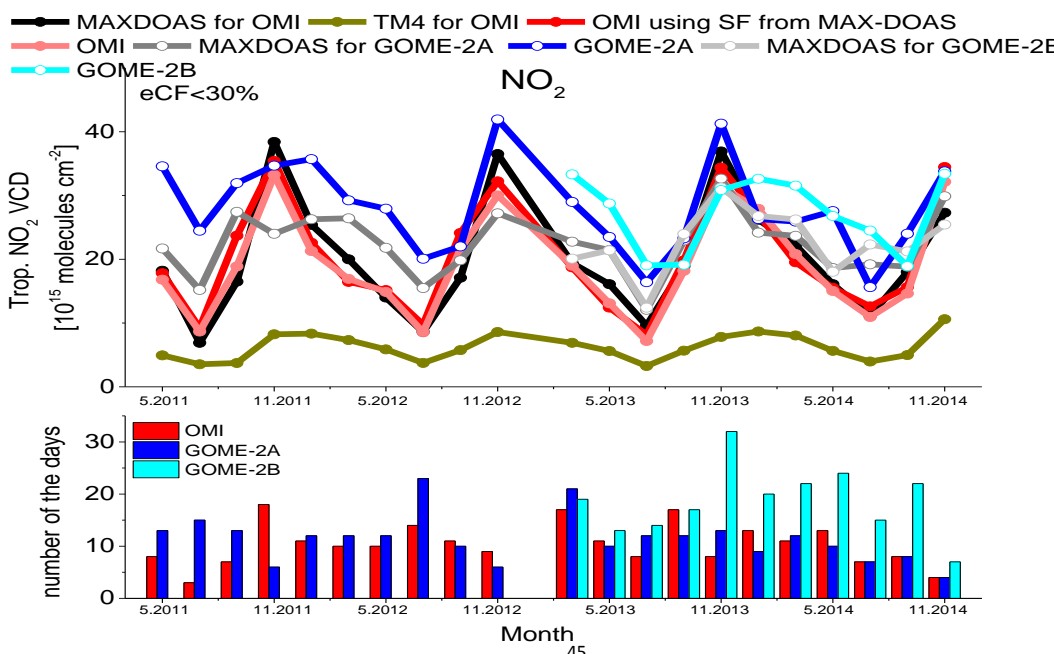



(b)

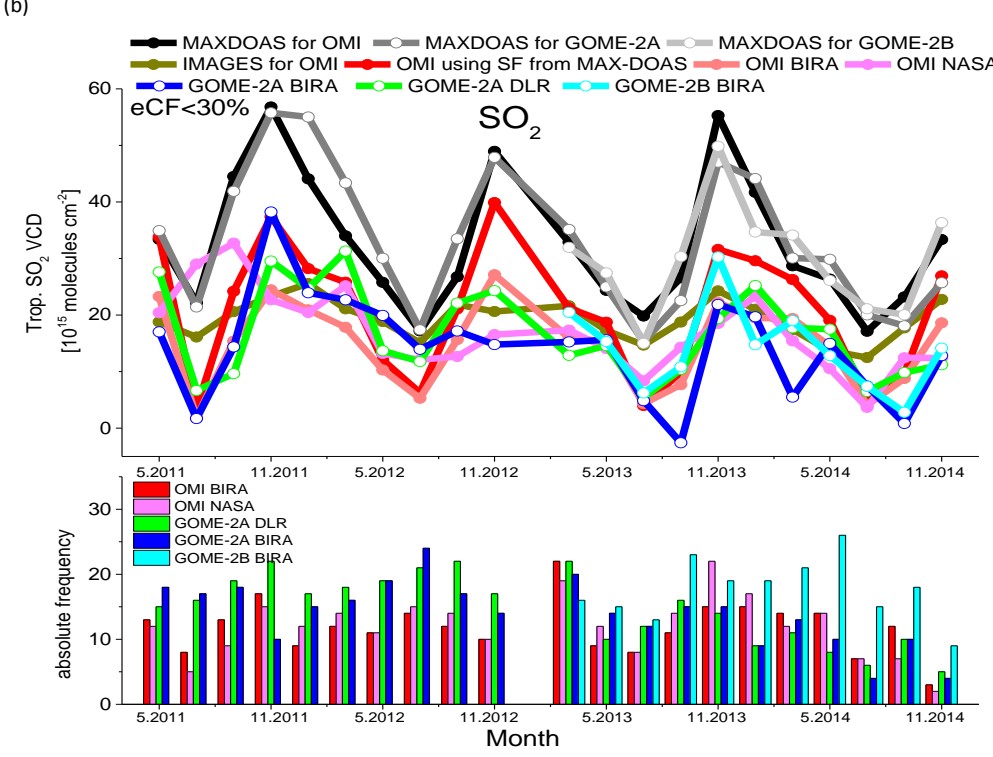

(c)

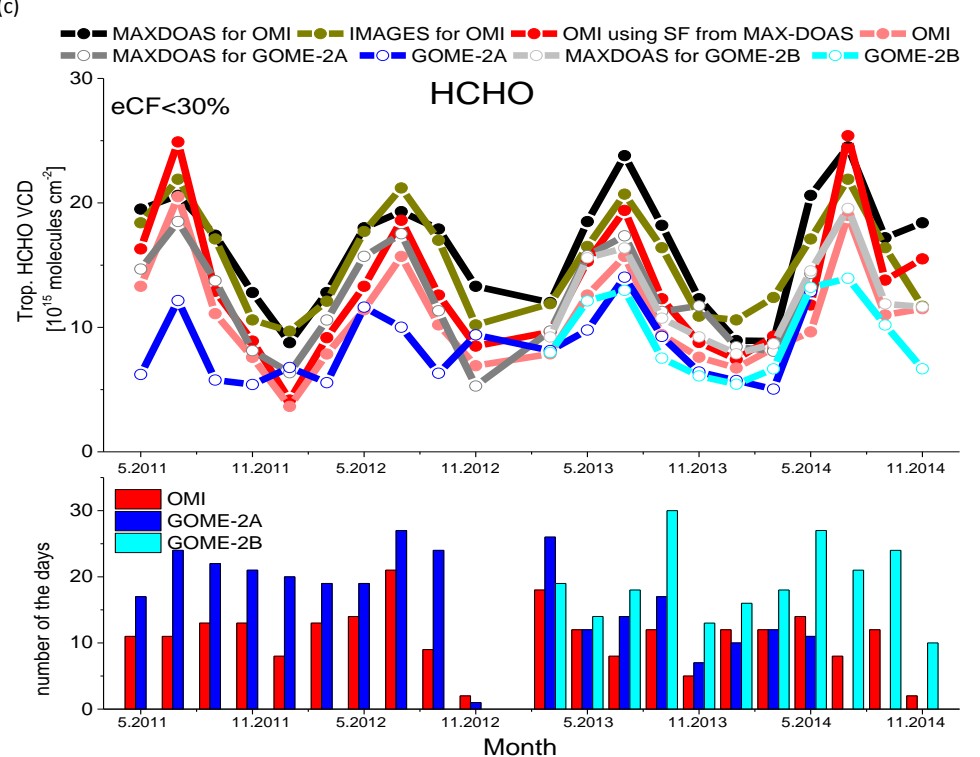



**Figure 17: Bi-monthly averaged tropospheric VCDs of NO₂ (a), SO₂ (b) and HCHO (c) derived from coincident satellite and MAX-DOAS observations for eCF <30%. Also shown are the corresponding CTM results (TM4 for NO₂, IMAGES for SO₂ and HCHO). In all subfigures the red and light red lines indicate the improved OMI tropospheric VCDs using the SFs from MAX-DOAS and the VCDs from the original OMI products, respectively. The numbers of the available days are shown in the bottom panel of each subfigure.**

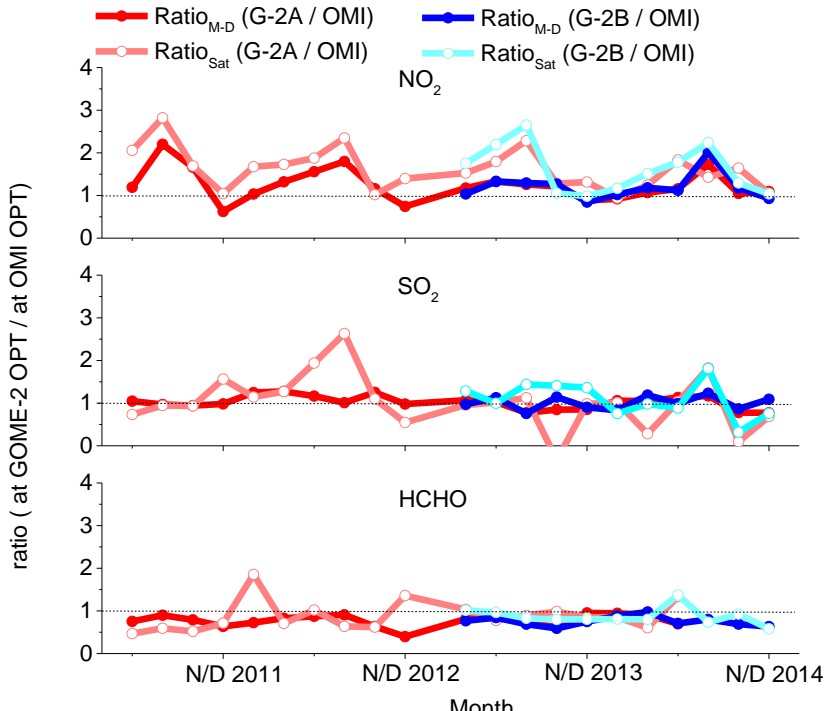

**Figure 18: Ratios between the bi-monthly mean tropospheric VCDs from GOME-2A\B and OMI (RatioSat) as well as the ratios between the corresponding MAX-DOAS observations (RatioM-D) for NO₂ (a), SO₂ (b) and HCHO (c), respectively. The light red (dark red) and light blue (dark blue) curves are corresponding to GOME-2A and GOME-2B results (coincident MAX-DOAS results with GOME-2A and GOME-2B), respectively. Note that for SO₂ the OMI and GOME-2A data from BIRA are used for the ratio calculations. The mean ratios for the shown data sets are presented in Table 1.**

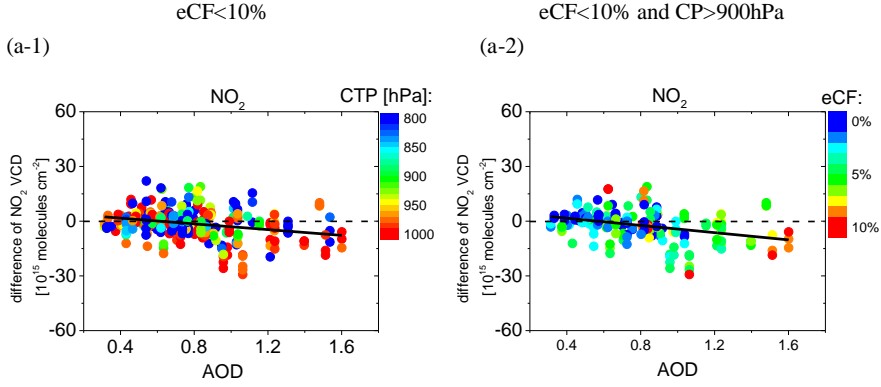



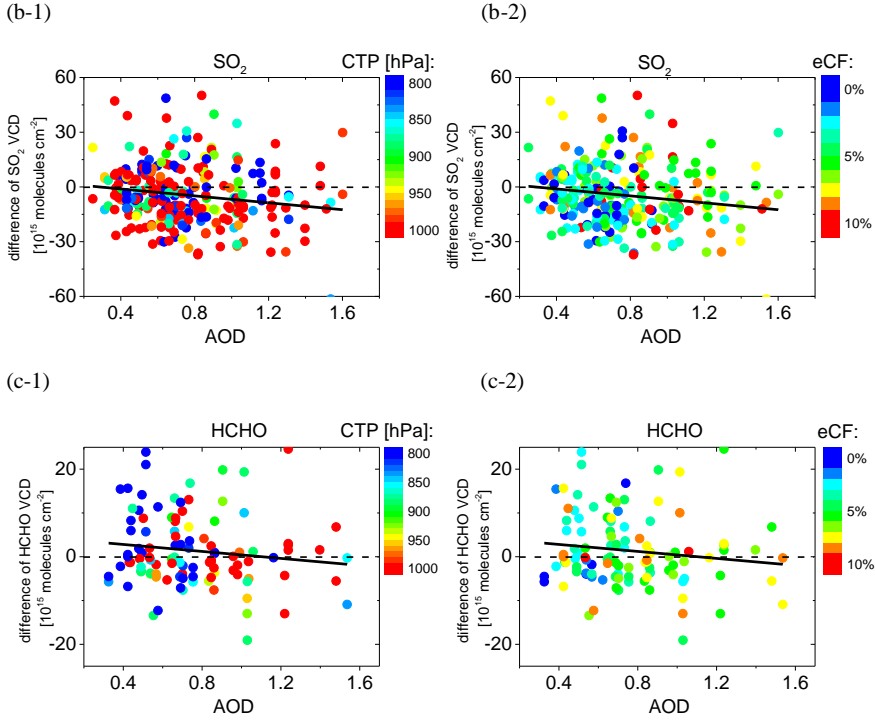

**Figure 19: Left panels: Differences of tropospheric VCDs of NO$_2$ (a-1), SO$_2$ (b-1) and HCHO (c-1) between for individual OMI observations (for eCF < 10%) and MAX-DOAS observations plotted against the AODs derived from the MAX-DOAS observations. Right panels: Same data as left, but observations with CTP<900hPa are skipped. The colours indicate the CTP (left) or eCF (right).**



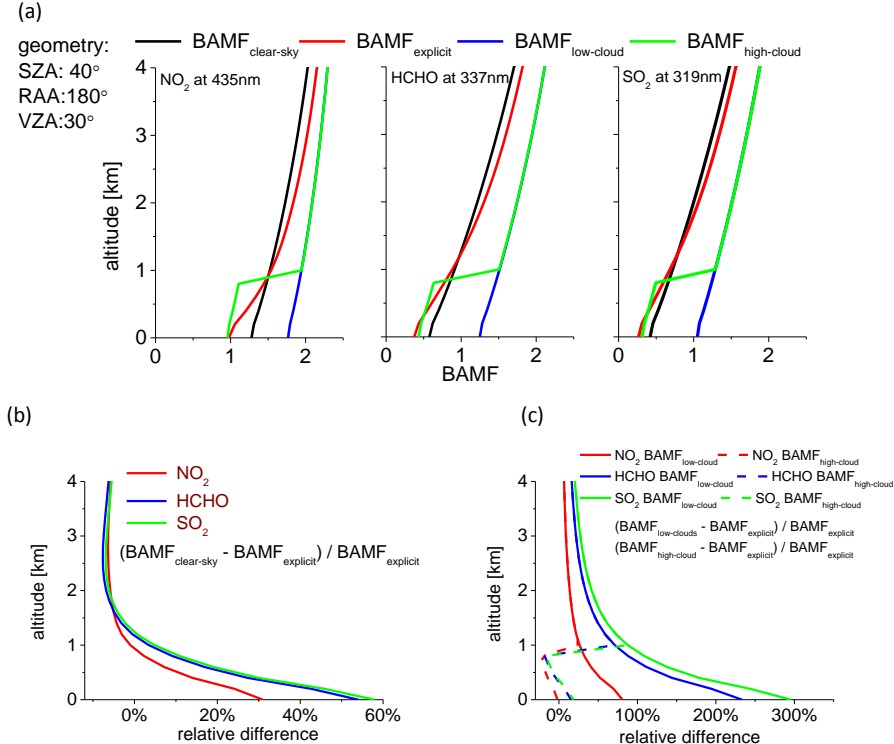

**Figure 20: (a) Simulated BAMF_clear-sky, BAMF_explicit, BAMF_low-cloud and BAMF_high-cloud of NO₂ at 435nm, HCHO at 337nm and SO₂ at 319nm for one typical nadir satellite observation (SZA of 40°, RAA of 180° and VZA of 30°). (b) Relative differences between BAMF_clear-sky and BAMF_explicit. (c) Relative differences between BAMF_low clouds or BAMF_high-clouds and BAMF_explicit. Note the different x-axes.**

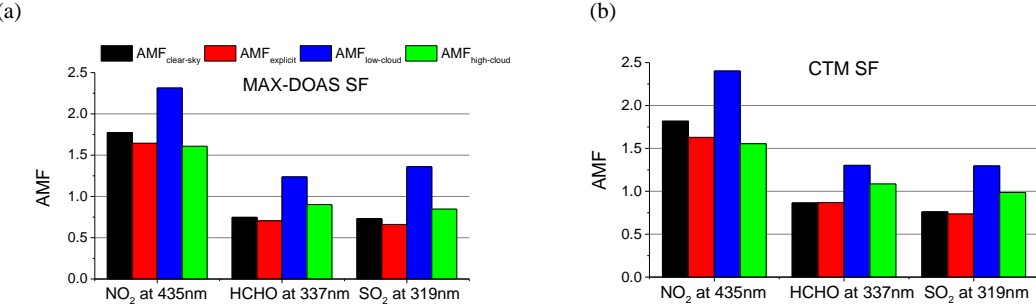

**Figure 21: AMFs calculated for different aerosol and cloud assumptions (for details see text) and different trace gases. The TG SFs are obtained from MAX-DOAS (a) or CTM (b), see also Fig. S5 in the Supplement.**





Table 1 Mean ratios for the data presented in Fig. 18.

|  | Ratio_M-D (G-2A / OMI) | Ratio_Sat (G-2A / OMI) | Ratio_M-D (G-2B / OMI) | Ratio_Sat (G-2B / OMI) |
|---|---|---|---|---|
| NO$_2$ | 1.25 | 1.62 | 1.20 | 1.61 |
| SO$_2$ | 1.02 | 1.02 | 1.01 | 1.09 |
| HCHO | 0.78 | 0.88 | 0.76 | 0.87 |

Table 2 Daily averaged AODs derived from MAX-DOAS observations, eCFs and CTPs derived from OMI for six cloud-free days with strong aerosol pollution.

| date | AOD from MAX-DOAS | OMI eCF [%] | OMI CTP [hPa] |
|---|---|---|---|
| Jan 26, 2012 | 0.56 | 9 | 955 |
| Oct 28, 2013 | 0.61 | 4 | 962 |
| Dec 10, 2011 | 0.69 | 5 | 830 |
| Nov 20, 2013 | 0.75 | 9 | 942 |
| Apr 22, 2012 | 0.85 | 6 | 991 |
| Nov 19, 2013 | 1.66 | 9 | 995 |