# Peer review of "Validation of OMI, GOME-2A, and GOME-2B tropospheric NO2, SO2, and HCHO products using MAX-DOAS observations from 2011 to 2014 in Wuxi, China: investigation of the effects of priori profiles and aerosols on the satellite products"

_Atmospheric Chemistry and Physics, 2016_

## Referee Comment (RC1) · Anonymous Referee #1 · 7 Oct 2016

In this paper, Wang et al. presented MAX-DOAS retrievals of NO2, SO2, and HCHO over Wuxi, a city within the heavily polluted Yangtze River Delta region in eastern China. They compared the MAX-DOAS retrievals with various OMI and GOME-2A/2B products. They also investigated the effects of a priori profiles and aerosols on satellite retrievals. The paper presents an interesting study that should be of interest to satellite trace gas retrieval community, especially the section discussing the effects of vertical profiles and aerosols on retrieval biases. The paper is generally well organized (given the multiple species/products/topics covered) and figures are mostly clear, although some improvement in writing would help. That said, I don't feel that the paper is quite

ready for publication in its current form. It is very long with a lot of long, complicated sentences that are not easy to understand. I also feel that some of the 21 multi-panel figures are not completely necessary and can be removed or moved to the supplemental material. Overall, I'd recommend that the authors try to make the paper more concise and focus more on the key points.

Specific comments: The introduction part may be a bit too long and can be shortened.

The authors used the entire section 3.1 and several figures (Fig. 2-6) in the main text to introduce how temporal/spatial averaging is done to match MAX-DOAS data with satellite data for the comparison. To me, such a lengthy discussion would be justified if the data averaging time and/or spatial averaging radius could be used for other validation/comparison studies. But I doubt that would be the case, given the location of the site and the inhomogeneous surface properties and trace gas loading over the area (Fig. 1). I feel that this section is probably best included in the supplementary material.

How was eCF calculated? And how was daily mean satellite VCDs calculated? Did the authors consider the size of the each satellite ground footprint?

Section 3.2: the NASA SO2 product essentially uses the same AMF for all pixels (regardless of viewing geometry and other conditions) that may lead to additional errors and affect its correlation with MAX-DOAS retrievals.

Section 3.2: Fig. 8, 10, 13, these figures may be replaced with a table.

Section 3.3: Fig. 14 – one would expect that the AMF calculated from TM4 profile would be on average smaller than AMF calculated using MAX-DOAS profile? Note that Fig. 14e and Fig. 15e show the same sign in AMF difference. The TM4 shape in Fig. 14a shows larger weight than MAX-DOAS in the lowest part of the profile, the IMAGES profile in Fig 15a, on the other hand, shows smaller weight than the MAX-DOAS profile in the in lowest layers.

Fig. 18, the authors may want to point out that GOME-2/OMI ratio for NO2 may be

[Figure]

much more meaningful than that for SO2, given the overall smaller retrieval uncertainty. Fig. 19: did the authors use MAX-DOAS profiles to correct for retrievals to isolate aerosols as a source of error in satellite retrievals?

Fig. 20 and section 3.6: can the authors specify the aerosol optical properties and size distribution assumed in the RTM calculations? Particularly, for the UV wavelengths especially 319 nm?

Section 4: instead of simply repeating the results already presented in the paper, the authors may consider condensing this part or provide some more in-depth discussion.
* * *

---

## Referee Comment (RC2) · Anonymous Referee #2 · 11 Oct 2016

Wang and co-authors investigate the quality of satellite retrievals of NO2, SO2, and HCHO over Wuxi in polluted China via a detailed comparison with ground-based column measurements obtained with the MAX-DOAS technique. This technique is sensitive to pollution in the lower atmosphere, and Wuxi in the Yangtze River area faces pervasive high levels of pollution from these gases and aerosols. The three years of MAX-DOAS measurements collected in Wuxi thus provide a very interesting data set to test the satellite retrievals, and provide guidance on how to use and possibly improve the retrievals. The authors report that the KNMI OMI NO2 (DOMINO v2) product agrees very well with the MAX-DOAS NO2 columns in Wuxi, especially in situations

with few clouds. But the KNMI NO2 products from the GOME-2 sensors tend to be overestimated. Because of this overestimation of GOME-2 NO2, also the satellite-derived NO2 diurnal cycle, while correct in sign, is overestimated. Satellite retrievals of SO2 and HCHO from BIRA and NASA tend to be underestimated by tens of per-cents relative to the MAX-DOAS measurements. These findings are relevant to the many users of satellite data interested in obtaining a better understanding of Chinese air pollution.

The paper then addresses some of the critical assumptions made in the satellite re-trievals on: the a priori trace gas vertical distribution in the retrievals, the cloud correc-tions made, the aerosol correction, and to what extent this proceeds implicitly via the cloud retrievals that are sensitive to aerosol effects (Leitao et al., 2011; Boersma et al., 2011; Castellanos et al., 2015; Chimot et al., 2016).

The comparison of MAX-DOAS and (CTM-derived) a priori profile shapes is a strong and innovative element of the study, and it is interesting to see how replacing the CTM-profiles with the actually observed profiles helps in improving the agreement between MAX-DOAS and satellite retrievals. Profile validation is dearly needed, and this study explores new avenues on how to achieve this, even though the vertical resolution of the MAX-DOAS and model profiles differ substantially. One highlight is that ∼20% of the SO2 and HCHO underestimation can be explained by the IMAGES profile shapes insuf-ficiently capturing the enhanced SO2 and HCHO concentrations in the Wuxi boundary layer.

Section 3.6 on aerosol effects on the AMFs is potentially also interesting, but I have serious concerns about the way it has been set up, and the current method does not allow drawing any firm conclusions. The section starts with an analysis of the NO2 discrepancies (satellite minus MAX-DOAS) as a function of AOD. This is relevant, but it does not become clear whether the discrepancies arise because of high AOD, because of residual clouds, or because of aerosols influencing the cloud fractions. Showing NO2 discrepancies only for cloud fraction < 0.1 is inconclusive since these cloud fractions

may represent real clouds, 'effective' clouds, or a combination of the two. To properly attribute the NO2 discrepancies to the effect of the aerosols, the authors should do what they did for Table 2: use MODIS to distinguish the cloud-free, aerosol loaded situations from the situations with residual clouds still present, and focus their analysis on that data cloud-cleared ensemble to rule out the contributions from clouds.

The subsequent box AMF calculations are only just a brief sensitivity study for a limited set of situations that is not representative for the large and robust data ensemble collected by the authors over Wuxi. For instance, only one viewing geometry has been tested (P18, L1). Furthermore, how much box AMFs differ between implicit and explicit aerosol corrections depends strongly on the exact assumption of AOT (profile), particle type, NO2 profile, albedo (why always 0.1?), as shown in many previous studies (e.g. Leitao et al. [2011]). None of this becomes clear on page 18, yet the conclusion is drawn that "the implicit aerosol correction typically causes larger bias of the satellite TG VCDs than the clear-sky assumption". This conclusion is based on only a few calculations that do not represent the full range of situations encountered by the retrievals under evaluation. The authors should have been as rigorous as in section 3.3 and replace the implicit aerosol correction by an explicit aerosol correction for the full set of satellite pixels.

The paper is too long. The section on the coincidence criteria can be shortened considerably. Other studies have investigated these issues, and the findings are probably specific for the Wuxi circumstances anyway. I recommend to move much of section 3.1, including the figures, to the supplementary material and focus on the final criteria, and then refer the reader for justification of these criteria to the supplement. Also sections 3.3 can be shortened; I'm not sure if for each retrieval the discrepancies as a function of cloud fraction need to be discussed (and shown) at length.

The systematic dependence of the HCHO spectral fitting uncertainty on the retrieved VCD for GOME-2 is intriguing, and deserves more attention. Why is this exactly? Why would this be different than for OMI? The authors should clarify these issues. Then

their decision to only validate OMI HCHO retrievals with fitting uncertainties < 7 1015 molec.cm-2 is questionable, since setting this threshold basically excludes half the data, not just some outliers or misfits. The authors may report that validation results for this sub-set are better than for the full set, as long as those results are also reported, because users of OMI HCHO data typically use all data, not just the sub-set retrieved with SCD uncertainties < 7 1015 molec.cm-2.

Specific comments

P3, L17-20: here it should be stressed that methodological assumptions on how clouds and aerosols should be accounted for in the AMF calculation matter, e.g. Lin et al. [2015].

P4, L1: studies investigating the shape factor are not "rare". There are many studies investigating the quality and effect of a priori profiles on retrievals and emission estimates; e.g. Boersma et al. [2004]; Hains et al., [2010]; Heckel et al. [2011]; Barkley et al. [2012]; Vinken et al. [2014]. Regardless, studying the impact of the shape factor remains relevant because profile measurements are indeed 'rare'.

P4, L35 and P5, L1-3: the argument in favour of the implicit aerosol correction in the Boersma-2011 paper is made for substantial AOD when particles are mostly scattering, i.e. not unlike cloud droplets. Castallanos et al. [2015] clearly showed that for absorbing particles and high AOD, the implicit aerosol correction breaks down. So the sentence that Castellanos demonstrated that for elevated biomass burning aerosols, the implicit correction does a good job is completely out of place. Their study showed that the implicit aerosol correction compares well with an explicit aerosol correction for low-modest AOD and SSA>0.95. For high AOD and lower SSA, the implicit aerosol correction breaks down, but these situations occur less frequently than the former.

P5, L31: it should be 'heavy fog'.

P6, L4-6: it should be clarified if the difference between the geometrical approximation and profile integration is systematic, or that the discrepancies are variable in both directions.

P6, L12: Capital S missing in 'sky'.

P6, L30: what is the source of information for the 68 x 14 km2 pixel size at OMI swath edges?

P7, L5-7: it would be appropriate to refer to Dirksen et al. [2011] here when discussing the data assimilation procedure to estimate the stratospheric background NO2. Similar to OMI SO2 from BIRA, DOMINO v2 can be regarded as the 'proxy' algorithm for the upcoming TROPOMI mission.

P8, L13: suggest to state 'similar data assimilation procedures'.

P9, L26: what is meant with the 'statistical uncertainty of the satellite data'?

P11, LL27-28:

P12, L11-14: with underestimations of ±50%, it is rather odd to conclude that GOME-2A products are "most accurate" for cloud fractions below 30%. Also the 'recommendation' to use SO2 observations with cloud fractions below 10% is far fetched. One might as well recommend to not use any SO2 data over the Yangtze area at all in view of the large, systematic biases shown in this study.

P12, L29: 'because of the weaker degradation' than OMI or GOME-2A? Please clarify.

P13, L1: dependencies.

P13, L4: when suggesting that HCHO products should be used for cloud fractions < 0.3, the authors should be more aware that their recommendation is based on the situation for Wuxi, which is not necessarily representative for situations with enhanced HCHO concentrations elsewhere (just think about the high aerosol loadings). Also, if they make such a recommendation, they should discuss it in the context of what the algorithm providers actually recommend for appropriate use of their data, and what has

typically been done in successful applications of the OMI HCHO data.

P14, L11: 'latitude range' should be altitude range, and 'larges' should be 'largest'.

P14, L11-14: it would be fair to clearly conclude here that the TM4 a priori profile shapes agree well with the MAX-DOAS NO2 profiles in an average sense.

P15, L15: please provide more detail on the months in the x-axis of Figure 17; we now only have tick marks for month 5 and 11. Some more specific indication for the bi-monthly averages would be useful.

P15, L23-24: please clarify why the TM4 NO2 columns are so much lower than those from the measurements. Later on page 16, same for SO2 modelled by IMAGES; why is HCHO from IMAGES doing a good job whereas SO2 is not?

P16, L22-24: it would be appropriate to refer to Boersma et al., JGR, 2008 here. That study was the first to investigate the diurnal cycle of NO2 with satellite measurements. Also some more explanation on what causes the diurnal changes in NO2, SO2, and HCHO columns is needed here.

P17, L18: some more information is needed on the 'clear-sky AMF' that is applied in SO2 and HCHO retrievals for cloud fractions < 0.1. How is such an AMF calculated – in an atmosphere with Rayleigh scattering only? Or is there some aerosol background assumed in the radiative transfer calculations?

P19, L24: please clarify what is meant with "cloud effects become significant". Do you mean that the discrepancies between MAX-DOAS and satellite columns are larger when cloud fractions are larger?

P19, L33-34: suggest to be more specific here and state that IMAGES profiles and TM4 profiles have been compared against MAX-DOAS profiles.

P20, L21-22: the sentence "NO2 satellite products systematically overestimate the magnitude of NO2 diurnal variation" is misleading. The diurnal variation is overestimated because the GOME-2 retrievals are too high, but OMI is in agreement with MAX-DOAS. Suggest to rephrase accordingly.

P20, L30-35: this part is too strong-worded and should be rephrased after the authors have addressed my concerns about section 3.6. The current sensitivity study provides too little ground to base these conclusions on.
* * *

---

## Author Response (AR1)

**Reply to Ref. #1**

First of all we want to thank this reviewer for the positive assessment of our manuscript and the constructive and helpful suggestions!

General comments:
In this paper, Wang et al. presented MAX-DOAS retrievals of NO2, SO2, and HCHO over Wuxi, a city within the heavily polluted Yangtze River Delta region in eastern China. They compared the MAX-DOAS retrievals with various OMI and GOME-2A/2B products. They also investigated the effects of a priori profiles and aerosols on satellite retrievals. The paper presents an interesting study that should be of interest to satellite trace gas retrieval community, especially the section discussing the effects of vertical profiles and aerosols on retrieval biases. The paper is generally well organized (given the multiple species/products/topics covered) and figures are mostly clear, although some improvement in writing would help. That said, I don't feel that the paper is quite ready for publication in its current form. It is very long with a lot of long, complicated sentences that are not easy to understand. I also feel that some of the 21 multi-panel figures are not completely necessary and can be removed or moved to the supple-mental material. Overall, I'd recommend that the authors try to make the paper more concise and focus more on the key points.

Author reply:
Many thanks for the positive assessment!
We made four important modifications to the paper. Firstly we moved section 3.1 about the coincidence criteria into supplement as section 1. Secondly we moved the section 2.1.2 about the cloud effects on MAX-DOAS observations into the supplement as section 2. Thirdly we rewrote the discussion on the aerosol effects in section 3.5 of the revised version (see general comment b from Reviewer 2). Fourthly we rewrote the discussion about the influence of the eCF on the shape factor effects on the AMF (section 3.2 of the revised version). This modification is following the specific comment #6.

**Specific Comments:**

1) The introduction part may be a bit too long and can be shortened.
Author reply:
We shortened the introduction by rewriting paragraphs 6 and 7 of the introduction section.

2) The authors used the entire section 3.1 and several figures (Fig. 2-6) in the main text to introduce how temporal/spatial averaging is done to match MAX-DOAS data with satellite data for the comparison. To me, such a lengthy discussion would be justified if the data averaging time and/or spatial averaging radius could be used for other validation/comparison studies. But I doubt that would be the case, given the location of the site and the inhomogeneous surface properties and trace gas loading over the area (Fig. 1). I feel that this section is probably best included in the supplementary material.
Author reply:
Thanks for the suggestion! We followed your suggestion to move the entire section 3.1 into the supplement. And we added a new paragraph in the beginning of section 3 to describe the main conclusions about the coincident criteria.

3) How was eCF calculated? And how was daily mean satellite VCDs calculated? Did the authors consider the size of the each satellite ground footprint?

Author reply:
The effective cloud fraction (eCF) is defined as in Stammes et al. (2008). We added this reference to section 2.2 of the revised version. We directly extract eCF data from the published operational products. We clarified the calculation of the daily and bi-monthly mean satellite VCDs in the beginning of section 3 as "Here it needs to be clarified that the daily and bi-monthly averaged satellite data are the averaged values of all satellite pixels located in the coincidence area around the measurement site (see below). The averaged MAX-DOAS data are the averaged values for all measurements within 2 hours around the satellite overpass time". Note that for the selection of the satellite data we don't explicitly use the size of the satellite ground footprint. Instead we use the distance of the center of the pixel to the measurement station. However, we chose distances according to the size of the footprint sizes and the expected gradients of the trace gases. Note that we also exclude the outermost pixels of the OMI swath (i.e. pixel numbers 1–5 and 56–60) as described at the end of section 2.2.

4) Section 3.2: the NASA SO2 product essentially uses the same AMF for all pixels (re-gardless of viewing geometry and other conditions) that may lead to additional errors and affect its correlation with MAX-DOAS retrievals.
Author reply:
Thanks for pointing out this important information! We agree that this simplified assumption might be part of the reason for the worse correlation of the NASA OMI product with MAX-DOAS compared to the BIRA product. We added this information in section 2.2 of the revised manuscript as "A fixed surface albedo (0.05), surface pressure (1013.25 hPa), solar zenith angle (30 °) and viewing zenith angle (0 °) as well as a fixed climatological $SO_2$ profile over the summertime eastern U.S. are assumed in the PCA retrieval (Krotkov et al., 2008).". It needs to be noted that the significantly worse $R^2$ for the OMI NASA product compared to the OMI BIRA product could partly be attributed to the assumed fixed measurement condition (and thus the fixed AMF) in the NASA PCA retrievals. However the similar slopes and MRDs between the two OMI products indicate that the simplification of the NASA PCA retrieval only slightly contributes to the systematic bias of the averaged values."

5) Section 3.2: Fig. 8, 10, 13, these figures may be replaced with a table.
Author reply:
Since figures represent the most direct way to show the dependence of the consistency on the eCF, we prefer to keep these figures.

6) Section 3.3: Fig. 14 – one would expect that the AMF calculated from TM4 profile would be on average smaller than AMF calculated using MAX-DOAS profile? Note that Fig. 14e and Fig. 15e show the same sign in AMF difference. The TM4 shape in Fig. 14a shows larger weight than MAX-DOAS in the lowest part of the profile, the IMAGES profile in Fig 15a, on the other hand, shows smaller weight than the MAX-DOAS profile in the in lowest layers.
Author reply:
Great thanks for pointing out this problem. The reviewer is correct with his/her description of the effect of the shape factor on the AMF. However because of the missing information on the shape factor above 4 km (which was not explicitly mentioned in the original manuscript) the SF effect

on the AMF was not correctly explained in the original manuscript. In the revised version (in section 3.2) we firstly clarified how we treat the concentration of TGs above 4km  as "It needs to be noted that only the profiles below 4km can be reliably drawn from MAX-DOAS observations. Thus the profile$_M$ between 4km and the tropopause (a fixed value of 16 km is used in this study) are derived from the corresponding CTM profiles of the individual satellite data sets. Therefore the SF$_M$ is derived from the combined profile$_M$ using Eq.3.". We also modified the explanation about the different effects on the AMF under different cloud conditions. For details, please see the revised manuscript.  In addition, in part 4 of section 3.2, we also point out that the lack of information about the profiles above 4km from MAX-DOAS observations is a potential error source in the analysis of SF effects on satellite AMF calculations.

7) Fig. 18, the authors may want to point out that GOME-2/OMI ratio for NO2 may be much more meaningful than that for SO2, given the overall smaller retrieval uncertainty.

Author reply:

Thanks for pointing out this issue!  We modified the description in section 3.4 in the revised version as "For NO$_2$, the Ratio$_{Sat}$ for both GOME-2 instruments show good agreement. Good agreement is also found for the seasonal variation with the MAX-DOAS results, but the absolute values differ. The systematic difference of Ratio$_{Sat}$ and Ratio$_{M-D}$ can be attributed to the known overestimation of the GOME-2 A/B tropospheric VCD compared to the MAX-DOAS results (see Fig. 12a). This finding also indicates that using GOME-2 and OMI data can lead to wrong conclusions about the diurnal cycles of NO$_2$. Also for the other trace gases we investigated the ratios between the different data sets. However, because of the larger uncertainties compared to NO$_2$, the conclusions for SO$_2$ and HCHO should be treated with care. For SO$_2$, although Ratio$_{Sat}$ shows several deviations from Ratio$_{M-D}$, Ratio$_{M-D}$ and Ratio$_{Sat}$ are consistent on average and close to unity during a whole year indicating similar SO$_2$ VCDs around the overpass times of GOME-2 and OMI. For HCHO, on average good agreement between Ratio$_{Sat}$ and Ratio$_{M-D}$ is found for GOME-2A and GOME-2B (except some outliers of Ratio$_{Sat}$). Interestingly, both Ratio$_{Sat}$ and Ratio$_{M-D}$ are below unity indicating lower HCHO VCDs in the morning than in the afternoon. ".

8) Fig. 19: did the authors use MAX-DOAS profiles to correct for retrievals to isolate aerosols as a source of error in satellite retrievals?

Author reply:

Yes, the OMI data used in Fig. 19 (Fig. 14 in the revised version) are the modified VCD using MAX-DOAS profiles. Thanks for pointing out the missing information. We clarified it in section 3.5 as "It needs to be noted that the OMI VCDs used in Fig. 14 are the modified values using the SFs derived from MAX-DOAS observations in order to isolate the aerosol effects.".

9) Fig. 20 and section 3.6: can the authors specify the aerosol optical properties and size distribution assumed in the RTM calculations? Particularly, for the UV wavelengths especially 319 nm?

Author reply:

We modified the description of aerosol properties used in the RTM simulations in section 3.5 of the revised version as "The aerosol optical properties (single scattering albedo of 0.9, asymmetry

parameter of 0.72, and Angstroem parameter of 0.85) are taken from the AERONET observations at the nearby Taihu station (Holben et al. 1998, 2001)." .

10) Section 4: instead of simply repeating the results already presented in the paper, the authors may consider condensing this part or provide some more in-depth discussion.
Author reply:
We made some modification in conclusion part to improve the discussion.

**Reply to Ref. #2**

First of all we want to thank this reviewer for the positive assessment of our manuscript and the constructive and helpful suggestions!

General comments
a) Wang and co-authors investigate the quality of satellite retrievals of NO2, SO2, and HCHO over Wuxi in polluted China via a detailed comparison with ground-based col-umn measurements obtained with the MAX-DOAS technique. This technique is sen-sitive to pollution in the lower atmosphere, and Wuxi in the Yangtze River area faces pervasive high levels of pollution from these gases and aerosols. The three years of MAX-DOAS measurements collected in Wuxi thus provide a very interesting data set to test the satellite retrievals, and provide guidance on how to use and possibly improve the retrievals. The authors report that the KNMI OMI NO2 (DOMINO v2) product agrees very well with the MAX-DOAS NO2 columns in Wuxi, especially in situations with few clouds. But the KNMI NO2 products from the GOME-2 sensors tend to be overestimated. Because of this overestimation of GOME-2 NO2, also the satellite-derived NO2 diurnal cycle, while correct in sign, is overestimated. Satellite retrievals of SO2 and HCHO from BIRA and NASA tend to be underestimated by tens of per-cents relative to the MAX-DOAS measurements. These findings are relevant to the many users of satellite data interested in obtaining a better understanding of Chinese air pollution.

The paper then addresses some of the critical assumptions made in the satellite re-trievals on: the a priori trace gas vertical distribution in the retrievals, the cloud correc-tions made, the aerosol correction, and to what extent this proceeds implicitly via the cloud retrievals that are sensitive to aerosol effects (Leitao et al., 2011; Boersma et al., 2011; Castellanos et al., 2015; Chimot et al., 2016). The comparison of MAX-DOAS and (CTM-derived) a priori profile shapes is a strong and innovative element of the study, and it is interesting to see how replacing the CTM-profiles with the actually observed profiles helps in improving the agreement between MAX-DOAS and satellite retrievals. Profile validation is dearly needed, and this study explores new avenues on how to achieve this, even though the vertical resolution of the MAX-DOAS and model profiles differ substantially. One highlight is that~20% of the SO2 and HCHO underestimation can be explained by the IMAGES profile shapes insufficiently capturing the enhanced SO2 and HCHO concentrations in the Wuxi boundary layer.

Author reply:
Many thanks for the positive assessment!
We made four important modifications for the paper. Firstly we moved section 3.1 about the coincidence criteria into supplement as section 1. Secondly we moved the section 2.1.2 about the cloud effects on MAX-DOAS observations into the supplement as section 2. Thirdly we rewrote the discussion on the aerosol effects in section 3.5 of the revised version (see general comment b). Fourthly we rewrote the discussion about the influence of the eCF on the shape factor effects on the AMF (section 3.2 of the revised version). This modification is following the specific comment #6 from Reviewer 1.

b) Section 3.6 on aerosol effects on the AMFs is potentially also interesting, but I have serious concerns about the way it has been set up, and the current method does not allow drawing any firm conclusions. The section starts with an analysis of the NO2 discrepancies (satellite minus MAX-DOAS) as a function of AOD. This is relevant, but it does not become clear whether the discrepancies arise because of high AOD, because of residual clouds, or because of aerosols influencing the cloud fractions. Showing NO2 discrepancies only for cloud fraction < 0.1 is inconclusive since these cloud

fractions. may represent real clouds, 'effective' clouds, or a combination of the two. To properly attribute the NO2 discrepancies to the effect of the aerosols, the authors should do what they did for Table 2: use MODIS to distinguish the cloud-free, aerosol loaded situations from the situations with residual clouds still present, and focus their analysis on that data cloud-cleared ensemble to rule out the contributions from clouds.

The subsequent box AMF calculations are only just a brief sensitivity study for a limited set of situations that is not representative for the large and robust data ensemble collected by the authors over Wuxi. For instance, only one viewing geometry has been tested (P18, L1). Furthermore, how much box AMFs differ between implicit and explicit aerosol corrections depends strongly on the exact assumption of AOT (profile), particle type, NO2 profile, albedo (why always 0.1?), as shown in many previous studies (e.g. Leitao et al. [2011]). None of this becomes clear on page 18, yet the conclusion is drawn that "the implicit aerosol correction typically causes larger bias of the satellite TG VCDs than the clear-sky assumption". This conclusion is based on only a few calculations that do not represent the full range of situations encountered by the retrievals under evaluation. The authors should have been as rigorous as in section 3.3 and replace the implicit aerosol correction by an explicit aerosol correction for the full set of satellite pixels.

Author reply:
Based on the comments of the reviewer, we rewrote the whole section 3.5 in the revised version about aerosol effects. One important point is that although the differences of clear sky AMF, implicit aerosol correction, and explicit aerosol correction have been systematically investigated in the previous studies (i.e. Leitão et al. (2010) and Chimot et al., 2016), here we characterize the aerosol effects for typical aerosol properties (profile, optical properties, and corresponding aerosol induced eCF and CTP) for a polluted region. Also, as mentioned by the reviewer the previous studies indicated that the aerosol effect "depends strongly on the exact assumption of AOT (profile), particle type, $NO_2$ profile, and albedo".

Thus we completely re-wrote the whole section, and we extended the RTM simulations to five different satellite observation geometries (listed in Table 2 of the manuscript) following the suggestions of the reviewer. The new results are shown in Fig. 17 of the revised manuscript. The new simulations indicate that the aerosol effects depends on the observation geometries, however the main conclusion on the effects of clear sky AMF and implicit aerosol corrections are consistent for different geometries.

For the discussion on Fig. 14 in the revised version (Fig. 19 in the original version) about the analysis of the $NO_2$ discrepancies as a function of AOD, we agree with the reviewer that for cloud fractions < 0.1 residual clouds can not certainly be excluded. Therefore we also used an additional criterium of CTP>900hPa, which can exclude residual cirrus clouds. Considering specific low altitude clouds (with either small OD and large geometric coverage or high OD and small geometric coverage) we performed additional simulation studies, which are described in the section 4 of the supplement. Our main conclusion is that for the selected cases the effect of residual clouds is negligible. We added this information to the main text of the manuscript.

In addition we excluded the part about the six pure aerosol pollution days, because we can not draw any general conclusion from these cases. We added a new figure (Fig. 15) showing aerosol-induced eCF and CTP derived from the OMI cloud retrieval as a function of the corresponding AOD derived from MAX-DOAS.

The reviewer asked the question "why always 0.1?". Here we updated the text as follows "The surface albedo is set to 0.1 for $NO_2$ and 0.05 for $SO_2$ and HCHO simulations based on the averaged value of the surface reflectivity data base derived from OMI by Kleipool et al. (2008) over Wuxi station." And we redo the RTM simulations with these surface albedo for more observation geometries using McArtim RTM.

c) The paper is too long. The section on the coincidence criteria can be shortened considerably. Other studies have investigated these issues, and the findings are probably specific for the Wuxi circumstances anyway. I recommend to move much of section 3.1, including the figures, to the supplementary material and focus on the final criteria, and then refer the reader for justification of

these criteria to the supplement. Also sections 3.3 can be shortened; I'm not sure if for each retrieval the discrepancies as a function of cloud fraction need to be discussed (and shown) at length.

Author reply:
Many thanks for the suggestion! We followed your suggestion to move the entire section 3.1 into the supplement. And we added a new paragraph at the beginning of section 3 to describe the main conclusions about the coincidence criteria.

d) The systematic dependence of the HCHO spectral fitting uncertainty on the retrieved VCD for GOME-2 is intriguing, and deserves more attention. Why is this exactly? Why would this be different than for OMI? The authors should clarify these issues. Then their decision to only validate OMI HCHO retrievals with fitting uncertainties $< 7 \ 10^{15}$ molec.cm$^{-2}$ is questionable, since setting this threshold basically excludes half the data, not just some outliers or misfits. The authors may report that validation results for this sub-set are better than for the full set, as long as those results are also reported, because users of OMI HCHO data typically use all data, not just the sub-set retrieved with SCD uncertainties $< 7 \ 10^{15}$ molec.cm$^{-2}$.

Author reply:
Unfortunately, at the moment we can't give any confirmed explanation on dependence of the HCHO spectral fitting uncertainty on the retrieved VCD for GOME-2 and the differences compared to OMI. We clarified this in the revised manuscript.

Concerning the filter of the fit error, the Fig. 6b in the revised version (Fig. 11 in the previous version) shows the comparisons of the linear regression parameters for the data before and after the filtering. We also add a new Fig. S12 in the supplement to show the effect of the fit error on the daily averaged data. The two comparisons demonstrate that the filter only considerably improves the correlation coefficient, but hardly changes the slopes and y-intercepts. Thus we conclude that it will not impact the conclusion on the systematic bias of the OMI HCHO products. The point is clarified in the revised manuscript. Furthermore, as mentioned in the paper, data with large uncertainty need to be excluded for a further investigation on cloud and aerosol effects. Otherwise the effects will be overwhelmed by the large uncertainties.

**Specific comments**

1) P3, L17-20: here it should be stressed that methodological assumptions on how clouds and aerosols should be accounted for in the AMF calculation matter, e.g. Lin et al. [2015].

Author reply:
We add this finding and the reference to the text.

2) P4, L1: studies investigating the shape factor are not "rare". There are many studies investigating the quality and effect of a priori profiles on retrievals and emission esti-mates; e.g. Boersma et al. [2004]; Hains et al., [2010]; Heckel et al. [2011]; Barkley et al. [2012]; Vinken et al. [2014]. Regardless, studying the impact of the shape factor remains relevant because profile measurements are indeed 'rare'.

Author reply:
We corrected the sentence as "Here it is important to note that many studies already investigated the quality and effect of a-priori SFs on satellite retrievals (i.e. Boersma et al., 2004; Hains et al., 2010; Heckel et al., 2011) and demonstrated that the SF effect on the tropospheric AMFs can dominate the systematic errors of tropospheric satellite products especially in highly polluted (especially urban and industrial) regions (Boersma et al., 2011, Theys et al., 2015 and De Smedt et al., 2015), Nevertheless, because profile measurements are rare, the SF effect is still not well understood in many regions."

3) P4, L35 and P5, L1-3: the argument in favour of the implicit aerosol correction in the Boersma-2011 paper is made for substantial AOD when particles are mostly scattering, i.e. not unlike cloud droplets. Castallanos et al. [2015] clearly showed that for absorbing particles and high AOD, the implicit aerosol correction breaks down. So the sentence that Castellanos demonstrated that for elevated biomass burning aerosols, the implicit correction does a good job is completely out of place. Their study showed that the implicit aerosol correction compares well with an explicit aerosol correction for low-modest AOD and SSA>0.95. For high AOD and lower SSA, the implicit aerosol correction breaks down, but these situations occur less frequently than the former.

Author reply:

Many thanks for this hint! We modified the sentence as "For mostly scattering aerosols at high altitudes the implicit aerosol correction can largely account for the aerosol effect on the TG products (Boersma et al., 2011). However in some important cases (for low altitude aerosols with high AOD and small SSA) the implicit correction might even increase the errors of the AMF Castellanos et al. (2015).".

4) P5, L31: it should be 'heavy fog'.

Author reply:
corrected

5) P6, L4-6: it should be clarified if the difference between the geometrical approximation and profile integration is systematic, or that the discrepancies are variable in both directions.

Author reply:

We clarified it as "Our previous study (Wang et al., 2016) demonstrated that the tropospheric trace gas VCDs from the full profile inversion are in general much more accurate than those from the geometric approximation. The discrepancy of VCDs between the two methods is systematic and can be mainly attributed to the errors of the geometric approximation, for which the errors can be up to 30% depending on the observation geometry, and the properties of aerosols and TGs. ".

6) P6, L12: Capital S missing in 'sky'.

Author reply:
Corrected.

7) P6, L30: what is the source of information for the 68 x 14 $km^2$ pixel size at OMI swath edges?

Author reply:

Many thanks for this hint! We changed the values to 150 x 13 $km^2$, see:
Levelt, P. F., van den Oord, G. H. J., Dobber, M. R., Malkki, A., Visser, H., de Vries, J., Stammes, P., Lundell, J., and Saari, H.: The Ozone Monitoring Instrument, IEEE Trans. Geosci. Remote Sens., 44, 1093–1101, 2006b.

8) P7, L5-7: it would be appropriate to refer to Dirksen et al. [2011] here when discussing the data assimilation procedure to estimate the stratospheric background $NO_2$. Similar to OMI $SO_2$ from BIRA, DOMINO v2 can be regarded as the 'proxy' algorithm for the upcoming TROPOMI mission.

Author reply:

We added the reference to Dirksen et al. [2011] and also clarified "The retrieval algorithm for DOMINO v2 forms the basis of $NO_2$ retrievals for the upcoming TROPOspheric Monitoring Instrument (TROPOMI) aboard the Sentinel-5 Precursor mission (Veefkind et al., 2012)."

9) P8, L13: suggest to state 'similar data assimilation procedures'.

Author reply:
Corrected.

10) P9, L26: what is meant with the 'statistical uncertainty of the satellite data'?

Author reply:
We delete "statistical".

11) P11, LL27-28

Author reply:

12) P12, L11-14: with underestimations of50%, it is rather odd to conclude that GOME- 2A products are "most accurate" for cloud fractions below 30%. Also the 'recommendation' to use SO2 observations with cloud fractions below 10% is far fetched. One might as well recommend to not use any SO2 data over the Yangtze area at all in view of the large, systematic biases shown in this study.

Author reply:
We modified the description as "Thus we conclude that the cloud effects on both GOME-2A products are appreciable for eCF > 30%. For the GOME-2B BIRA data, an obvious decrease of $R^2$ and slope is found for eCF > 10%, while for eCF>30% largely variable MRDs are found. Thus clouds can considerably impact the GOME-2B BIRA product for eCF > 10%, and more significantly for eCF > 30%. ".

13) P12, L29: 'because of the weaker degradation' than OMI or GOME-2A? Please clarify.

Author reply:
We changed the text to "because of the weaker degradation of GOME-2B during the short time after launch compared to OMI and GOME-2A.".

14) P13, L1: dependencies.

Author reply:
Corrected.

15) P13, L4: when suggesting that HCHO products should be used for cloud fractions < 0.3, the authors should be more aware that their recommendation is based on the situation for Wuxi, which is not necessarily representative for situations with enhanced HCHO concentrations elsewhere (just think about the high aerosol loadings). Also, if they make such a recommendation, they should discuss it in the context of what the algorithm providers actually recommend for appropriate use of their data, and what has typically been done in successful applications of the OMI HCHO data.

Author reply:
We modified the sentence as "In general cloud effects on the HCHO products become substantial for eCF > 30% for the three satellite instruments. However it needs to be noted that our findings are derived for one location (Wuxi) and might not be fully representative for other locations. The use of the HCHO products with eCF < 40% is recommended by the retrieval algorithm developer (De Smedt et al., 2015).".

16) P14, L11: 'latitude range' should be altitude range, and 'larges' should be 'largest'.

Author reply:
Corrected.

17) P14, L11-14: it would be fair to clearly conclude here that the TM4 a priori profile shapes agree well with the MAX-DOAS NO2 profiles in an average sense.

Author reply:
We added this finding.

18) P15, L15: please provide more detail on the months in the x-axis of Figure 17; we now only have tick marks for month 5 and 11. Some more specific indication for the bi-monthly averages would be useful.

Author reply:
We modified the figure accordingly. Note that the Fig. 17 in the previous version is Fig. 12 in the revised version.

19) P15, L23-24: please clarify why the TM4 NO2 columns are so much lower than those from the measurements. Later on page 16, same for SO2 modelled by IMAGES; why is HCHO from IMAGES doing a good job whereas SO2 is not?

Author reply:
We added that "The significant underestimation of the TM4 $NO_2$ VCDs could be due to many factors, most importantly the limited spatial model resolution, which is especially relevant for species with strong horizontal gradients such as $NO_2$ and $SO_2$ (see Figure 1), but also possible errors in the emissions, transport and/or chemical mechanism. The determination of the specific contributions of the different error sources should be the subject of future studies." in the revised version. We also mention that the results of the IMAGES model for $SO_2$ and HCHO need further investigations in the future.

20) P16, L22-24: it would be appropriate to refer to Boersma et al., JGR, 2008 here. That study was the first to investigate the diurnal cycle of NO2 with satellite measurements. Also some more explanation on what causes the diurnal changes in NO2, SO2, and HCHO columns is needed here.

Author reply:
We added the reference and now mention that "The diurnal variations can be attributed to the complex interaction of the primary and secondary emission sources, depositions, atmospheric chemical reactions, and transport processes.".

21) P17, L18: some more information is needed on the 'clear-sky AMF' that is applied in SO2 and HCHO retrievals for cloud fractions < 0.1. How is such an AMF calculated – in an atmosphere with Rayleigh scattering only? Or is there some aerosol background assumed in the radiative transfer calculations?
Author reply:
'clear-sky AMF' means in an atmosphere with Rayleigh scattering only. We clarified this in the revised version.

22) P19, L24: please clarify what is meant with "cloud effects become significant". Do you mean that the discrepancies between MAX-DOAS and satellite columns are larger when cloud fractions are larger?

Author reply:
Yes. We already clarified it in the sentence before that sentence as "The consistency (correlations and systematic bias) of satellite data with MAX-DOAS results deteriorates with increasing eCF.".

23) P19, L33-34: suggest to be more specific here and state that IMAGES profiles and TM4 profiles have been compared against MAX-DOAS profiles.

Author reply:
We added this information in the revised version.

24) P20, L21-22: the sentence "NO2 satellite products systematically overestimate the magnitude of NO2 diurnal variation" is misleading. The diurnal variation is overestimated because the GOME-2 retrievals are too high, but OMI is in agreement with MAX-DOAS. Suggest to rephrase accordingly.

Author reply:
We changed the text to "The systematic difference of RatioSat and RatioM-D can be attributed to the known overestimation of the GOME-2 A/B tropospheric VCD compared to the MAX-DOAS results (see Fig. 12a). This finding also indicates that using GOME-2 and OMI data can lead to wrong conclusions about the diurnal cycles of $NO_2$.".

25) P20, L30-35: this part is too strong-worded and should be rephrased after the authors have addressed my concerns about section 3.6. The current sensitivity study provides too little ground to base these conclusions on.

Author reply:
We modified the section 3.5 in the revised version (section 3.6 in the previous version). Thus the relevant conclusion part is re-written as:
Finally we studied aerosol effects on the OMI products over Wuxi station based on the MAX-DOAS observations. We find that the underestimation of the TG VCDs derived from satellite observations for mainly cloud-free observations compared to the MAX-DOAS observations systematically increases with AOD. We also investigate the aerosol effect based on RTM simulations. Here it is also possible to separate the aerosol effect into two contributions: a) the effect of using a clear sky AMF instead of an AMF taking explicitly into account the aerosol effects, and b) the effect of aerosols on the cloud retrievals, which are used in the satellite TG retrievals (implicit aerosol correction). We find that for the measurements affected by high aerosol loads in Wuxi, in general the effect of the implicit cloud correction on the retrieved TG VCDs is much stronger than the difference of a clear sky AMF compared to an AMF taking explicitly into account the aerosol extinction. We also showed that for eCF <10% and CTP >900hPa the effect of residual clouds can be neglected if aerosol extinction is explicitly taken into account. Moreover, the observed underestimation of the OMI $NO_2$ VCD for large AOD can be well explained by the error caused by the implicit aerosol correction. Therefore it could be reasonable to apply the clear-sky AMFs in the satellite retrievals of TG tropospheric VCDs in case of CTP > 900hPa and eCF<10% if explicit aerosol information is not available.

**Validation of OMI, GOME-2A, and GOME-2B tropospheric NO$_2$, SO$_2$, and HCHO products using MAX-DOAS observations from 2011 to 2014 in Wuxi, China: investigation of the effects of  priori profiles  and aerosols  on the satellite products**

Yang Wang[1], Steffen Beirle[1], Johannes Lampel[1,2], Mariliza Koukouli[3], Isabelle De Smedt[4], Nicolas Theys[4], Ang Li[5], Dexia Wu[5], Pinhua Xie[5,6,7], Cheng Liu[8,6,5], Michel Van Roozendael[4], Trissevgeni Stavrakou[4], Jean-François Müller[4], and Thomas Wagner[1]

[1] Max Planck Institute for Chemistry, Mainz, Germany

[2] Institute of Environmental Physics, University of Heidelberg, Heidelberg, Germany

[3] Laboratory of Atmospheric Physics, Aristotle University of Thessaloniki, Thessaloniki, Greece

[4] Belgian Institute for Space Aeronomy (BIRA-IASB), Brussels, Belgium

[5] Anhui Institute of Optics and Fine Mechanics, Chinese Academy of Sciences, Hefei, China

[6] CAS Center for Excellence in Urban Atmospheric Environment, Institute of Urban Environment, Chinese Academy of Sciences, Xiamen, China

[7] School of Environmental Science and Optoelectronic Technology, University of Science and Technology of China, Hefei, China

[8] School of Earth and Space Sciences, University of Science and Technology of China, Hefei, China

*Correspondence to*: Yang Wang (y.wang@mpic.de), Ang Li (Angli@aiofm.ac.cn), Cheng Liu (chliu81@ustc.edu.cn)

**Abstract.**

Tropospheric vertical column densities (VCDs) of NO$_2$, SO$_2$, and HCHO derived from Ozone Monitoring Instrument (OMI) on AURA and Global Ozone Monitoring Experiment 2 aboard METOP-A (GOME-2A) and METOP-B (GOME-2B) are widely used to characterize the global distributions, trends, dominating sources of the trace gases and for comparisons with chemical transport models (CTM). We use tropospheric VCDs and vertical profiles of NO$_2$, SO$_2$ and HCHO derived from MAX-DOAS measurements from 2011 to 2014 in Wuxi, China, to validate the corresponding products derived from OMI and GOME-2A/B by different scientific teams (daily and bimonthly averaged data). Prior to the comparison,  the spatial and temporal coincidence criteria for MAX-DOAS and satellite data  are determined by a  sensitivity study using different spatial and temporal averaging conditions. Cloud effects  on both MAX-DOAS and satellite observations are also investigated. Our results indicate that the discrepancies between satellite and MAX-DOAS results increase with increasing effective cloud fractions and are dominated by the cloud effect on the satellite products.  In comparison with MAX-DOAS, we found  a systematic underestimation of all SO$_2$ (40% to 57%) and HCHO products (about 20%) and an overestimation of the GOME-2A/B NO$_2$ products (about 30%)

consistent DOMINO version 2 NO$_2$ product. To better understand the reasons for the differences, we evaluated the a-priori profile shapes  used in OMI retrievals (derived from CTM) by comparison with those derived from MAX-DOAS observations. Significant differences are  found for SO$_2$ and HCHO profile shapes derived from the IMAGES model, whereas on average good agreement was found for the NO$_2$ profile shapes derived from the TM4 model  The recalculated satellite VCDs agree better with the MAX-by a use ofderived those from products The improvement is strongest for periods with large trace gas VCDs. Finally~~ Furthermore, we investigate the effect of aerosols on the satellite retrievals. For OMI observations of NO$_2$, a systematic underestimation is found for a large AOD, which is mainly attributed to effect of the aerosols on the cloud retrieval and subsequent application of a cloud correction scheme (implicit aerosol correction). In contrast, the effect of aerosols on the clear sky AMF (explicit aerosol correction) has a smaller effect. For SO$_2$ and HCHO observations selected in the same way, no clear aerosol effect is found, probably because for the considered data no cloud correction is applied (but also because of the larger scatter). From our findings we conclude that for ~~for the common aerosol and trace gas scenarios in Wuxi. For the scenarios, the implicit correction (with corresponding effective cloud fraction (eCF) below 10% and cloud top pressure (CTP) of 1000 to 850 hPa) could cause a larger negative bias of tropospheric VCDs than the clear sky assumption compared to explicit aerosol corrections. And the bias can amount to up to about 45%, 15% and 35% for NO$_2$, HCHO, and SO$_2$, respectively~~ We find an increasing underestimation of the OMI NO$_2$, SO$_2$ and HCHO products with increasing AOD by up to 8%, 12% and 2%, respectively. One reason for this finding is that aerosols systematically affect the satellite cloud retrievals and can lead to apparent effective cloud fractions of up to 10% and apparent cloud top pressures of down to 830 hPa for the typical urban region in Wuxi. We show that in such cases the implicit aerosol correction could cause a strong underestimation of tropospheric VCDs by up to about 45%, 77% and 100% for NO$_2$, SO$_2$ and HCHO, respectively..Therefore it could be reasonable to apply the clear sky AMFs in the retrievals of TG tropospheric VCDsin caseofIn additionin comparisons with MAX-DOAS, the diurnal variations (ratios) ofFor 
[revised manuscript text omitted]
.~~, reported that the aerosol profiles over Wuxi station representing a box-like shape near the surface and an exponential decrease above 0.5 to 1 km. About 70% of aerosols accumulate in air layers below 1 km in general. The averaged profile derived from all the measurements under cloud free sky conditions are shown in supplementary Fig. S19. In order to evaluate the average and maximum aerosol effects, the averaged AOD of 0.8 and typical high AOD of 1.5 are used for the explicit aerosols in the following RTM simulations. The same profile shape with the averaged aerosol profile, which is shown in the supplementary Fig. S19, is used for the two aerosol scenarios.profile shapes (shape factors (SFs)) of As height profiles of wandunder cloud free sky conditions are used in the following simulations to calculate AMFs.~~ 
[revised manuscript text omitted]

~~The averaged effect is evaluated based on a typical scenario of aerosols and TGs in a specific location. Therefore we firstly characterized the aerosol and TG scenarios as well as the corresponding aerosol induced OMI eCF and CTP. Secondly the two types of aerosol effects are evaluated by a use of RTM simulations for different satellite observation geometries. Our results indicate that the implicit correction generally cause a larger negative bias of tropospheric VCDs than the clear sky assumption for the typical aerosol scenario in Wuxi, especially in case of CTP > 900hPa. The error of the implicit aerosol correction depends on the profiles of aerosols and TGs, observation geometries, and satellite cloud products. And the dependence of the underestimation of 
[revised manuscript text omitted]

**3. Comparisons of AMFs  for aerosols and  low clouds**

 In this section we performed McArtim RTM simulations to estimate the effect of low level clouds on the TG AMFs compared to TG AMFs for aerosols . In the simulation the aerosol properties are assumed to be the same as in section 3.5 of the manuscript (two scenarios with either AOD of 0.8 or 1.5). The cloud properties are chosen to obtain the same radiance and O₄ SCDs (at 477nm) as for the aerosol scenarios, but the  SSA is set to 1 and the asymmetry parameter g to 0.85 . The simulations are performed for  five satellite observation geometries as listed in Table 2 in the main manuscript.  We chose two different cloud types: a) a homogeneous clouds with low optical depth  covering the entire  satellite pixel, and b) an optically thick cloud covering only 20% of the  satellite pixel . It should be noted that the

cloud extinction profiles can not unambiguously be determined based on the radiance and $O_4$ SCDs only. Thus we assumed two different types of clouds, which represent the most extreme cases:  one case is a 'near-surface clouds' with a constant extinction starting from surface. For this cloud type the  cloud top height is then  derived based on $O_4$ SCDs; another cloud type is a 'lifted clouds' with the fixed vertical extension of 400 m (constant extinction in the cloud layer). For this cloud type  the height of the cloud center is  derived from the $O_4$ SCDs. Thus, in total all simulations are performed for four types of clouds, which are referred to as 'near-surface homogeneous clouds', 'near-surface partial clouds', 'lifted homogeneous clouds', and 'lifted partial clouds'.. The comparisons of the corresponding sun-normalized radiances and $O_4$ SCDs between  the aerosol profiles and with the derived cloud profiles are shown in Fig. S8 and S9, respectively.  The derived cloud extinction profiles are shown in Fig. S10. Here it should be noted that the simulations for the near-surface homogeneous clouds can not match the $O_4$ SCD  derived for the aerosols with AOD of 1.5. Finally the AMFs for the derived clouds are compared with those for the corresponding aerosols for the different observation geometries. The corresponding relative differences  are shown in Fig. S11. In general the differences are smaller than 10% for $NO_2$, and 5% for $SO_2$ and HCHO, except for the g3 geometry, for which the $NO_2$ AMF difference amounts up to 20%. Thus we conclude that in general the influence of residual clouds on the satellite TG retrievals is of importance. Here it should be noted that especially over polluted regions situations with eCF<10% and CTP > 900 hPa typically represent cases with aerosol pollution. If in addition residual clouds are present, they typically co-exist with high aerosol amounts. Thus our simulation results (based on pure cloudy cases without aerosols) represent not typical but rather extreme cases.

(a)

[Figure]

(b)

[Figure]

AOD of 1.5 at 360nm

**Figure S8: sun-normalized radiances simulated by RTM at 477nm for the two aerosol cases and the different clouds scenarios shown in Fig. S10. The different labels at the x-axis indicate  different observation geometries (see Table 2 in the main text). The radiances for the  AOD of 0.8  are shown in (a) and those for AOD of 1.5  in (b).**

(a)

[Figure]

AOD of 0.8 at 360nm

(b)

[Figure]

**Figure S9: O$_4$ SCDs simulated by  at 477nm for the two aerosol cases and the different clouds scenarios shown in Fig. S10 The different labels at the x-axis indicate different observation geometries  (see Table 2 in the main text). The radiances for the  AOD of 0.8  are shown in (a) and those for AOD of 1.5  in (b).**

(a)                                                                                          (b)

[Figure]

(c)                                                                                          (d)

[Figure]

**Figure S10:** Derived cloud extinction profiles for the four cloud types and viewing geometries the black curves indicate the aerosol extinction profiles, the coloured lines the cloud extinction profiles. Homogeneous clouds and partial clouds (see text) are shown in (a) and (b). The solid and dashed curves indicate the near-surface and lifted clouds, respectively. (a) and (b) are for the aerosol of AOD of 0.8; (c) and (d) are for AOD of 1.5.

(a)

[Figure]

(b)

[Figure]

**Figure S11: Relative differences of the three TG AMFs  for the four cloud types  and  the corresponding aerosol profiles. The different labels at the x-axis indicate different observation geometries  (see Table 2). The MAX-DOAS and CTM SFs are used for the calculations shown in the left and right columns. The aerosol profiles of AOD of 0.8 and 1.5 are used in subfigure (a) and (b), respectively.**

**4 Other figures**

(a)                                                    (b)

[Figure]

**Figure S12: daily averaged HCHO tropospheric VCD derived from OMI observations are plotted against those derived from MAX-DOAS observations for eCF<30%. And linear regressions are also shown. The OMI data before and after the filter of VCD fit error < 7×10$^{15}$ molecules cm$^{-2}$ are plotted in subplot (a) and (b), respectively.**

(a)  OMI                                                    (b) GOME-2A

[Figure]

(c) GOME-2B

[Figure]

**Figure S13: HCHO tropospheric VCDs for OMI pixels for eCF<30% are plotted against those derived from MAX-DOAS observations with the color map of eCF; the linear regression parameters are acquired for eCF<30% and for eCF<10%, respectively. (b) Scattered plots are same as in (a)**

(a)                                                         (b)

[Figure]

[Figure]

**Figure S14: (a) Averaged difference between the NO$_2$ SF from CTM (SF$_C$) and from MAX-DOAS (SF$_M$) for different eCF bins in the altitude range of 4km to 16km. (b) Averaged NO$_2$ BAMF for satellite observation for different eCF bins in the altitude range of 4km to 16km. (c) NO$_2$ tropospheric AMFs calculated with averaged SF$_M$ (marked by "M") and SF$_C$ (marked by "C"), respectively; the partial AMFs below and above 4km are marked by green and blue columns, respectively.**

(a)                                                      (b)

[Figure]

(c)

[Figure]

**Figure S15: Same as Fig. S14, but for SO$_2$.**

(a)
[Figure]
 (b)

[Figure]

(c)

[Figure]

**Figure S16: Same as Fig. S14, but for HCHO.**

[Figure]

(c) GOME-2B

[Figure]

(a)

[Figure]

(b)

[Figure]

(c)

[Figure]

**Figure S3S17: For eCF<30%, weekly cycles of VCDs of NO₂ (a), SO₂ (b) and HCHO (c) derived from different satellite instruments, corresponding coincident MAX-DOAS measurements. In all the subfigures the red and light red lines indicate the improved OMI tropospheric VCDs using the SFs from MAX-DOAS and the original VCDs from OMI products, respectively. The numbers of the available days in each two-month bin from different satellite products are shown in the bottom of each subfigure.**

[Figure]

**Figure S18: eCF and CTP from individual OMI observations are plotted against AOD around 360nm derived from AERONET Taihu station during the whole measurement in the condition of eCF<10% and CTP > 900hPa. The red bars on the right and bottom indicate the frequency of eCF, CTP, and AOD in different value intervals. The red lines are the linear regressions of the scatter plots. The correlation coefficients are shown in the plots. The color map in (a) and (b) indicate CTP and eCF, respectively.**

[Figure]

**Figure S19: Averaged aerosol extinction profiles and SF of NO$_2$, SO$_2$ and HCHO derived from all MAX-DOAS measurements under cloud-free sky conditions. The dashed curves indicate the corresponding averaged SF derived from CTM simulations for NO$_2$ (TM4), SO$_2$ (IMAGES) and HCHO (IMAGES).** (a)

Jan. 26, 2012          Oct. 28, 2013          Dec. 10, 2011

[Figure]

Nov. 20, 2013          Apr. 22, 2012          Nov. 13, 2013

(b)  Jan. 26, 2012                                        Oct. 28, 2013

Dec. 10, 2011                                        Nov. 20, 2013

[Figure]

Apr. 22, 2012                                                    Nov. 13, 2013

[Figure]

**Figure S4: (a) visual images from MODIS on the Aqua satellite on the six days with strong aerosol pollutants, obtained from the MODIS Rapid Response website, NASA/GSFC (http://aeronet.gsfc.nasa.gov/cgi-bin/bamgomas_interactive); (b) AODs from MAX-DOAS and the nearby Taihu AERONET station on the six days.**

[Figure]

Figure S5: Averaged aerosol extinction profiles and SF of NO₂, SO₂ and HCHO derived from all MAX-DOAS measurements under cloud-free sky conditions. The dashed curves indicate the corresponding averaged SF derived from CTM simulations for NO₂ (TM4), SO₂ (IMAGES) and HCHO (IMAGES).

---

## Author Response (AR2)

**Reply to Co-Editor**

Dear Dr. Bryan N. Duncan

We submitted the revised version of our paper.

We carefully improved the writing of the manuscript as suggested by the reviewer. We also added the requested information about the wavelengths of the sun photometer.

Please note that additional language corrections will be applied by the editorial office.

With best regards,

Yang Wang

**Reply to Ref. #1**

First of all we want to thank this reviewer for the positive assessment of our manuscript and the constructive and helpful suggestion!

-Would suggest the authors to try to further improve the writing. Also the authors should add a note that the RTM simulations use AERONET retrieved aerosol properties. But AERONET measurements do not extend to the wavelengths for $SO_2$ retrievals (~319 nm in this paper).

Reply: We modified the paper and give the exact wavelength of the aerosol properties derived from the AERONET observations. Single scattering albedo and asymmetry parameter are at 438nm. Angstroem parameter is 
[revised manuscript text omitted]